# How Do Large Language Monkeys Get Their Power (Laws)?

**Rylan Schaeffer** [1]  **Joshua Kazdan** [2]  **John Hughes** [3][4]  **Jordan Juravsky** [1]  **Sara Price** [4]  **Aengus Lynch** [4][5]
**Erik Jones** [6]  **Robert Kirk** [5]  **Azalia Mirhoseini** [1]  **Sanmi Koyejo** [1]

## Abstract

Recent research across mathematical problem solving, proof assistant programming and multimodal jailbreaking documents a striking finding: when (multimodal) language model tackle a suite of tasks with multiple attempts per task – succeeding if any attempt is correct – then the negative log of the average success rate scales a power law in the number of attempts. In this work, we identify an apparent puzzle: a simple mathematical calculation predicts that on each problem, the failure rate should fall exponentially with the number of attempts. We confirm this prediction empirically, raising a question: from where does aggregate polynomial scaling emerge? We then answer this question by demonstrating per-problem exponential scaling can be made consistent with aggregate polynomial scaling if the distribution of single-attempt success probabilities is heavy tailed such that a small fraction of tasks with extremely low success probabilities collectively warp the aggregate success trend into a power law - even as each problem scales exponentially on its own. We further demonstrate that this distributional perspective explains previously observed deviations from power law scaling, and provides a simple method for forecasting the power law exponent with an order of magnitude lower relative error, or equivalently, $\sim 2 - 4$ orders of magnitude less inference compute. Overall, our work contributes to a better understanding of how neural language model performance improves with scaling inference compute and the development of scaling-predictable evaluations of (multimodal) language models.

[1]Stanford Computer Science [2]Stanford Statistics [3]Speechmatics [4]ML Alignment & Theory Scholars [5]University College London [6]Anthropic. Correspondence to: Rylan Schaeffer <rschaef@cs.stanford.edu>, Sanmi Koyejo <sanmi@cs.stanford.edu>.

*Proceedings of the $42^{nd}$ International Conference on Machine Learning*, Vancouver, Canada. PMLR 267, 2025. Copyright 2025 by the author(s).

## 1. Introduction

Scaling behaviors of large neural language models have surprised and fascinated engineers, scientists and society alike (Hestness et al., 2017; Kaplan et al., 2020; Brown et al., 2020a; Hoffmann et al., 2022; Ganguli et al., 2022; Sorscher et al., 2022; Wei et al., 2022b; Schaeffer et al., 2023; OpenAI et al., 2024), shaping engineering, economic and governmental interests in frontier AI systems (Bommasani et al., 2021; Eloundou et al., 2023; Anderljung et al., 2023; Wang et al., 2023; Reuel et al., 2024; Besiroglu et al., 2024a; Maslej et al., 2024). For a more thorough exposition of relevant literature, please see Related Work (Section 6).

One direction of renewed interest is inference-time compute scaling, whereby compute is controllably increased at inference to improve the performance of a model, e.g., Pachocki et al. (2024). In this direction, recent research discovered that language model success rates scale predictably with the number of independent attempts made at accomplishing a task. Specifically, in a paper titled, "Large Language Monkeys: Scaling Inference Compute with Repeated Sampling," Brown et al. (2024) studied how language model performance changes at mathematical problem solving and coding problems when $k$ independent attempts are sampled per problem. Performance on the $i$-th problem was measured using the expected (over attempts) success rate (Kulal et al., 2019; Chen et al., 2021), defined as:

$$\text{pass}_i@\text{k} \stackrel{\text{def}}{=} \mathop{\mathbb{E}}_{k \text{ Attempts}} \Big[ \mathbb{I}[\text{Any attempt on } i\text{-th problem succeeds}] \Big]. \quad (1)$$

Using the unbiased and numerically stable estimator of Chen et al. (2021) (for details, see Appendix B), Brown et al. (2024) found that the negative log averaged-over-$P$-problems success rate falls as a power law with the number of independent attempts per problem $k$:

$$-\log \left( \frac{1}{P} \sum_{i=1}^{P} \text{pass}_i@\text{k} \right) \approx ak^{-b}, \quad (2)$$

for model-specific and benchmark-specific constants $a, b > 0$ (Fig. 1 Top). Soon after, on a separate topic of jailbreaking multimodal language models via text, image and audio

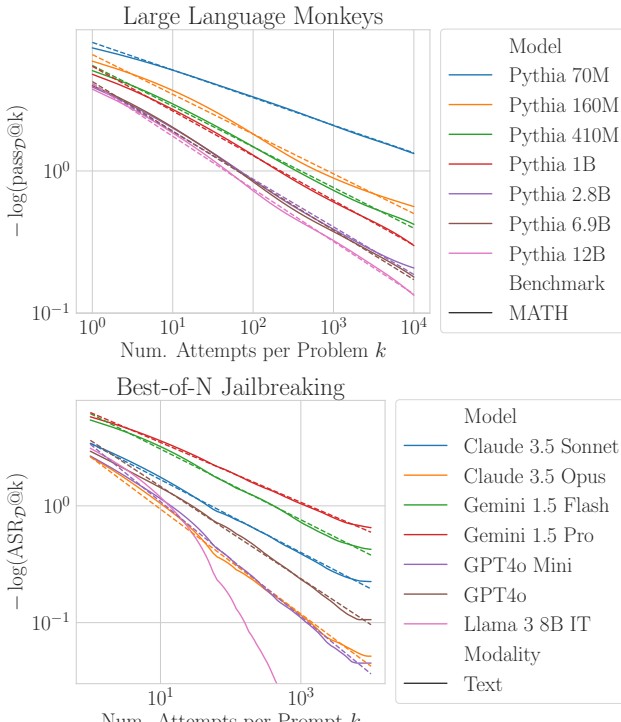

**Figure 1: Power Law Scaling in Language Models from Repeat Sampling.** Top: Brown et al. (2024) found the negative log average pass rate $-\log(\text{pass}_{\mathcal{D}}@\text{k})$ at solving mathematical problems scales polynomially (i.e., as a power law) with the number of independent attempts per problem $k$. Bottom: Hughes et al. (2024) similarly found the negative log average attack success rate $-\log(\text{ASR}_{\mathcal{D}}@\text{k})$ when jailbreaking multimodal language models scales polynomially with the number of jailbreak attempts per prompt. Should such power law scaling be expected? From where do large language monkeys obtain their power (laws)?

attacks, independent work by Hughes et al. (2024) studied jailbreaking success rates when $k$ independent attempts are made per harmful prompt. Performance was measured using Attack Success Rate (ASR) at $k$:

$$\text{ASR}_{\text{i}}@\text{k} \overset{\text{def}}{=}$$
$$\underset{k \text{ Attempts}}{\mathbb{E}} \Big[ \mathbb{I}[\text{Any attack on } i\text{-th prompt succeeds}] \Big]. \quad (3)$$

This "Best-of-N Jailbreaking" attack similarly discovered that the negative log averaged-over-$P$-prompts attack success rate fell as a power law with the number of jailbreak attempts per prompt $k$:

$$-\log\left(\frac{1}{P}\sum_{i=1}^{P}\text{ASR}_{\text{i}}@\text{k}\right) \approx ak^{-b}, \quad (4)$$

for model-specific and modality-specific constants $a, b > 0$ (Fig. 1 Bottom). For the specific coefficients from both

papers, see Appendix. C. As a minor matter of terminology, both papers frame their results in terms of "coverage" – the fraction of problems that can be solved after $k$ attempts per problem – but as Brown et al. (2024) pointed out, coverage is equivalent to the average success rate (Appendix D); we prefer this latter framing as it avoids the binary implication that each problem either is or is not solved after $k$ attempts.

## 2. Should Power Law Scaling Be Expected?

Should we expect large language monkeys to have such power (laws)? That is, should the negative log of the average success rate scale polynomially with the number of independent attempts $k$? As we now explain mathematically and demonstrate empirically, such polynomial scaling with $k$ is perhaps surprising because, for any single problem, the negative log success rate at $k$ should fall exponentially with $k$; the intuition is that $\text{pass}_{\text{i}}@\text{k}$ is 1 unless *all* attempts fail, and since attempts are independent, the probability that all fail is exponentially unlikely with the number of attempts.

Mathematically, on any given attempt, the model has probability $\text{pass}_{\text{i}}@1$ of solving the $i$-th problem. Recalling that $\text{pass}_{\text{i}}@\text{k}$ is defined as 1 if *any* of the $k$ attempts succeed, 0 otherwise, by linearity of expectation and by independence of the $k$ attempts, we can rewrite $\text{pass}_{\text{i}}@\text{k}$ as:

$$\text{pass}_{\text{i}}@\text{k} = \underset{k \text{ Attempts}}{\mathbb{E}} \Big[ 1 - \mathbb{I}[\text{All } k \text{ Attempts Fail}] \Big] \quad (5)$$

$$= 1 - \prod_{j=1}^{k} \underset{1 \text{ Attempt}}{\mathbb{E}} \Big[ \mathbb{I}[j\text{-th Attempt Fails}] \Big]. \quad (6)$$

The probability that the $j$-th attempt fails is one minus the probability that the $j$-th attempt succeeds. Since each attempt is i.i.d. with success probability $\text{pass}_{\text{i}}@1$, we find

$$\text{pass}_{\text{i}}@\text{k} = 1 - (1 - \text{pass}_{\text{i}}@1)^k. \quad (7)$$

For large $k$, $(1 - \text{pass}_{\text{i}}@1)^k$ will be small. Recalling that the Taylor Series expansion of $\log(1 + x)$ for small $x$ is $\sum_{i=1}^{\infty}(-1)^{i-1}x^i/i \approx x$, we have:

$$-\log(\text{pass}_{\text{i}}@\text{k}) = -\log\left(1 - (1 - \text{pass}@1)^k\right) \quad (8)$$

$$\approx (1 - \text{pass}_{\text{i}}@1)^k. \quad (9)$$

Thus, *for any single problem*, we should expect the negative log expected (over attempts) success rate to fall *exponentially* with $k$, not polynomially with $k$.

To confirm this claim, we plotted the scaling of model performance on each problem – measured either by $-\log(\text{pass}_{\text{i}}@\text{k})$ or by $-\log(\text{ASR}_{\text{i}}@\text{k})$ – against the number of independent attempts $k$. We specifically used Brown

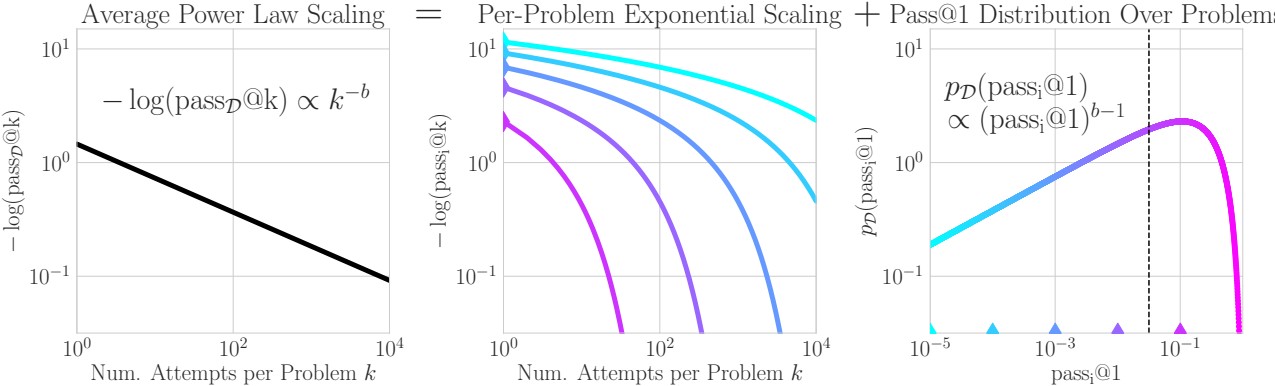

Figure 2: **Schematic: The Origin of Power Laws from Scaling Inference Compute via Repeat Sampling.** The $-\log(\text{pass}_\mathcal{D}@\text{k})$ scales as a power law with the number of attempts per problem $k$ (left). This arises from a combination of two factors: (1) for each problem, $-\log(\text{pass}_i@\text{k})$ scales exponentially with $k$ (center), and (2) the distribution (over problems in the dataset) of single-attempt success rates $\text{pass}_i@1$ itself has a left power-law tail of small values (right).

et al. (2024)'s data of the Pythia language model family (Biderman et al., 2023) solving 128 mathematical problems from MATH Hendrycks et al. (2021) as well as Hughes et al. (2024)'s data from jailbreaking frontier AI systems – Claude, GPT4 (OpenAI et al., 2024), Gemini (Team et al., 2024a;b) and Llama 3 8B Instruction Tuned (IT) (Grattafiori et al., 2024) – on 159 prompts from HarmBench (Mazeika et al., 2024). For each individual mathematical problem and jailbreaking prompt, we found the negative log expected (over attempts) success rates fall exponentially with $k$ as expected (Fig. 3), including on Llama 3 8B IT which does not exhibit an aggregate power law (Fig. 1).

## 3. Distribution of Per-Problem Single-Attempt Success Rates Creates Power Law Scaling

How does polynomial scaling of the negative log *average* success rate emerge from exponential scaling of the negative log *per-problem* success rate? The answer to this question *must* lie in the distribution $\mathcal{D}$ over benchmark problems of single attempt (i.e., $k = 1$) success rates because this distribution's density $p_\mathcal{D}(\text{pass}_i@1)$ links the per-problem scaling behavior to the aggregate scaling behavior via the definition of the aggregate success rate $\text{pass}_\mathcal{D}@\text{k}$:

$$\text{pass}_\mathcal{D}@\text{k} \overset{\text{def}}{=} \underset{\text{pass}_i@1 \sim \mathcal{D}}{\mathbb{E}} \left[ \text{pass}_i@\text{k}(\text{pass}_i@1) \right]$$
$$= 1 - \int_0^1 (1 - \text{pass}_i@1)^k \, p_\mathcal{D}(\text{pass}_i@1) \, d\,\text{pass}_i@1 .$$
$$(10)$$

Based on a known result that power laws can originate from an appropriately weighted sum of exponential functions (Appendix E.1), we begin by considering simple distributions for the single-attempt success probabilities and asking which yield power law scaling between $-\log(\text{pass}_\mathcal{D}@\text{k})$

and $k$, as well as what properties of the distributions set the scaling exponent. In Appendices E.3-E.8, we derive that several simple distributions yield power law scaling with different exponents whereas others do not:

$$-\log\left(\text{pass}_{\text{Uniform}(0,\,\beta\leq 1)}@\text{k}\right) \propto k^{-1}.$$
$$-\log\left(\text{pass}_{\text{Beta}(\alpha,\beta)}@\text{k}\right) \propto k^{-\alpha}.$$
$$-\log\left(\text{pass}_{\text{Kumaraswamy}(\alpha,\beta)}@\text{k}\right) \propto k^{-\alpha}.$$
$$-\log\left(\text{pass}_{\text{ContinuousBernoulli}(\lambda < 1/2)}@\text{k}\right) \propto k^{-1}.$$
$$-\log\left(\text{pass}_{\text{Reciprocal}(0<\alpha<\beta<1)}@\text{k}\right) \propto \frac{(1-\alpha)^k}{k}.$$

To test this understanding, we examined whether the data of Brown et al. (2024) and Hughes et al. (2024) had per-problem single-attempt success rate distributions that matched one of these simple distributions (Fig. 4). We found that the distributions could indeed be well fit by a 3-parameter $\text{Kumaraswamy}(\alpha, \beta, a = 0, c)$ distribution with scale parameter $c$ (Fig. 4, black dashed lines); we found the scale parameter was critical to obtain good fits because the standard 2-parameter Kumaraswamy distribution is supported on $(0, 1)$ whereas most single-attempt success distributions have a smaller maximum such as 0.01 or 0.1.

More generally, what are the distributional properties that create such power law scaling and that set the specific power law exponent? As we now show, the negative log average success rate will exhibit power law scaling in $k$ with exponent $b$ if and only if the distribution over problems of single-attempt success probabilities itself behaves like a power law near 0 with exponent $b - 1$:

**Theorem 3.1** (Sufficiency of Power-Law Left Tail in Dis-

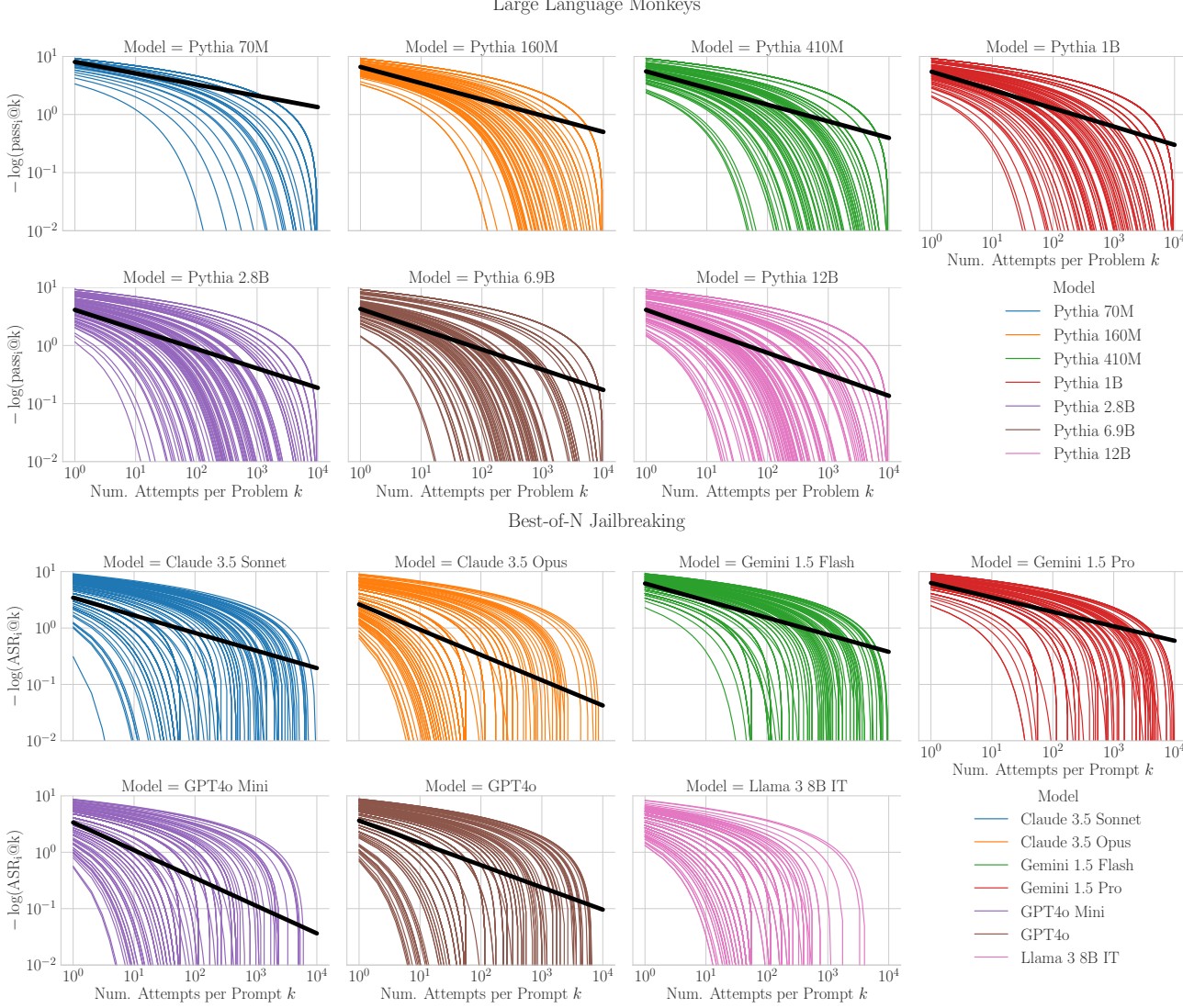

Figure 3: **Per-problem performance scales exponentially with the number of attempts per problem** $k$. Top: Pythia language models on 128 problems from MATH, with performance on the $i$-th problem measured as $-\log(\text{pass}_i@\text{k})$. Bottom: Frontier AI models on jailbreaking prompts from HarmBench, with performance on the $i$-th problem measured as $-\log(\text{ASR}_i@\text{k})$. In both settings, on each problem, the negative log *per-problem* success rate falls exponentially with the number of independent attempts $k$. However, the negative log *average* success rate falls as a power law with $k$ (black).

tribution of Single-Attempt Success Rates). *Let $\mathcal{D}$ be a probability distribution on $[0, 1]$ with PDF $p_{\mathcal{D}}(\text{pass}_i@1)$. Suppose there exist constants $b > 0$, $C > 0$, $\theta > 0$ and $\delta > 0$ such that, for all $0 < \text{pass}_i@1 < \delta$, we have*

$$p_{\mathcal{D}}(\text{pass}_i@1) \ = \ C \cdot (\text{pass}_i@1)^{b-1} + O\big((\text{pass}_i@1)^{b-1+\theta}\big).$$

*Then, for large $k$,*

$$-\log\big(\text{pass}_{\mathcal{D}}@\text{k}\big) \ \sim \ C\,\Gamma(b)\,k^{-b}.$$

**Theorem 3.2** (Necessity of Power-Law Left Tail in Distribution of Single-Attempt Success Rates). *Let $\mathcal{D}$ be a*

*distribution over $\text{pass}_i@1 \in [0, 1]$ with PDF $p_{\mathcal{D}}(\text{pass}_i@1)$. Suppose there exist constants $b > 0$ and $A > 0$ such that for large $k$,*

$$-\log\big(\text{pass}_{\mathcal{D}}@\text{k}\big) \sim A\,k^{-b}.$$

*Then, under mild regularity assumptions, the probability density must satisfy*

$$p_{\mathcal{D}}(\text{pass}_i@1) \ \sim \ \frac{A}{\Gamma(b)}\,(\text{pass}_i@1)^{b-1} \quad as \ \text{pass}_i@1 \to 0^+.$$

In Fig. 2, we illustrate this connection schematically. For proofs, see Appendices E.8 and E.9. These results clarify

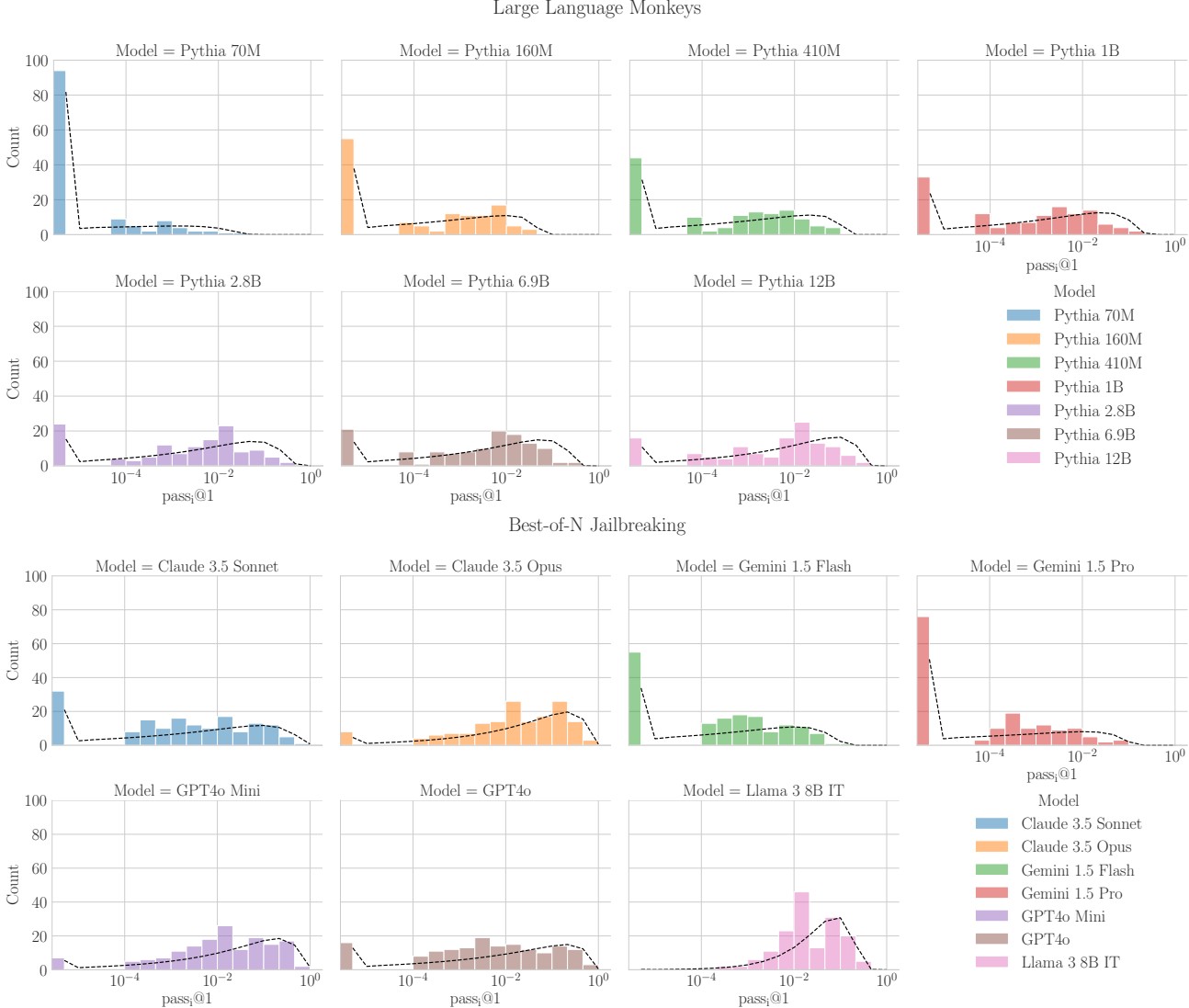

Figure 4: **Single-Attempt Success Rates Distributions Possess Power Law-Like Left Tails.** Pythia language models on 128 MATH problems (top) and frontier AI systems on 159 HarmBench prompts (bottom) exhibit distributions (over problems) of $\text{pass}_i@1$ and $\text{ASR}_i@1$ with power law-like tails that are well fit by scaled Beta-Binomial distributions (black dashed lines), which produce aggregate power law scaling. Note that Llama 3 8B Instruction Tuned (IT) does not possess a power law tail, explaining why the model did not exhibit aggregate power law scaling under Best-of-N jailbreaking (Sec. 4).

that whenever $-\log(\text{pass}_\mathcal{D}@\text{k})$ exhibits power-law decay in $k$ with exponent $b$, the distribution over problems of single-attempt success rates *must* have "polynomial weight" near $\text{pass}_i@1 = 0$, i.e. $p_\mathcal{D}(p) = \Theta(p^{b-1})$.

To offer intuition, we know that each problem is being solved by the model (or equivalently, each prompt is jailbreaking the model) exponentially quickly. If one looks across all problems in the benchmark, some have $\text{pass}_i@1$ so small that they remain unsolved for many, many attempts. Whether these "tiny-$\text{pass}_i@1$" problems still matter at large $k$ depends on how *many* such problems there are. Polyno-

mial density near 0 "piles up" enough hard problems in just the right way such that even though each of those problems is being solved exponentially quickly, the *aggregate* success rate over problems decreases at only a power-law rate in $k$. A more succinct mathematical summary is that, for a compound binomial distribution, the lower tail probability controls the upper tail of the marginal survivor function.

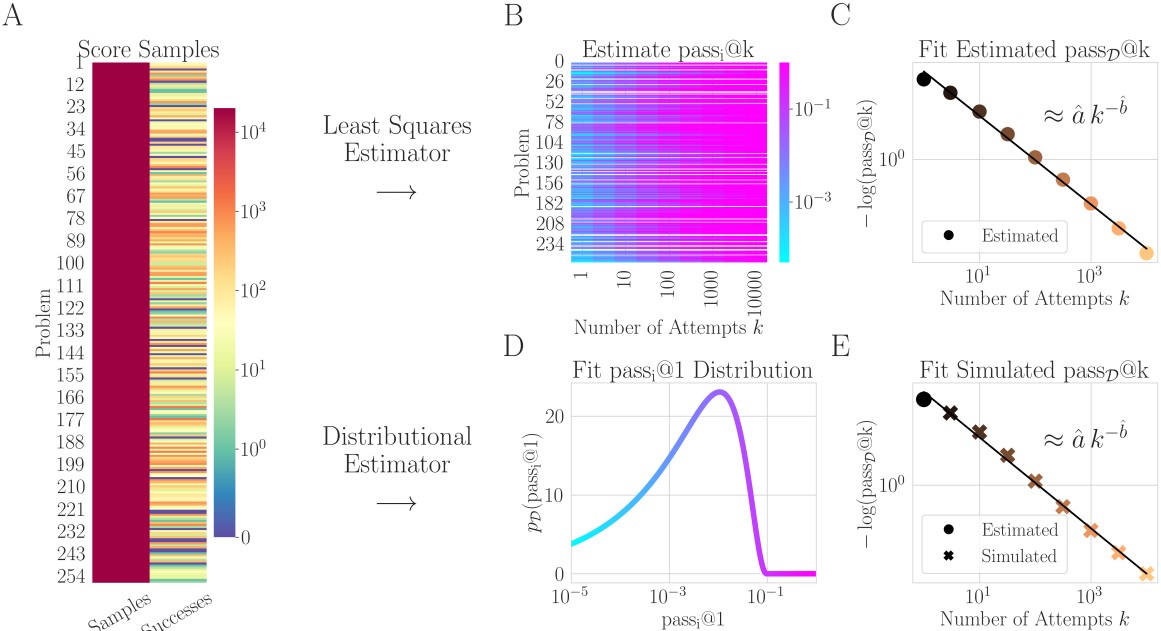

Figure 5: **Schematic: Two Estimators of Power Law Parameters for Scaling Inference Compute via Repeat Sampling.** (A) Both estimators begin by generating many samples per prompt, then computing the number of successes per prompt. In the standard least squares power law parameter estimator (top), (B) $\text{pass}_i@k$ is estimated for each $i$-th problem at multiple $k$ values, then (C) averaged over problems and fit with linear regression in log-log space. In the distributional power law parameter estimator (bottom), (D) a distribution $\mathcal{D}$ is fit to estimates of $\text{pass}_i@1$, then (E) the single-attempt success probability distribution is used to simulate $\text{pass}_{\mathcal{D}}@k$ at arbitrary $k$ values for linear regression in log-log space.

## 4. Lack of Distributional Structure Explains Deviations from Power Law Scaling

Notably, previous papers observed that not every model exhibits power law scaling in every setting. To highlight one, Hughes et al. (2024) observed that when jailbreaking Meta's Llama 3 8B Instruction Tuned (IT) model (Grattafiori et al., 2024), the $-\log(\text{ASR}_{\mathcal{D}}@k)$ fell faster than any power law (Fig. 1), i.e., the $\text{ASR}_{\mathcal{D}}@k$ rose much more quickly than the other frontier AI systems. Based on our mathematical insights and the empirical per-problem single-attempt attack success rates (Fig. 4), we can understand why: Llama 3 8B IT could be successfully jailbroken on every prompt within the permitted sampling budget and thus had no heavy left tail necessary to create the aggregate power law scaling.

## 5. A New Distributional Estimator for Predicting Power Law Scaling

A natural consequence of this connection between the scaling of $-\log(\text{pass}_{\mathcal{D}}@k)$ and the left tail of the distribution $p_{\mathcal{D}}(\text{pass}_i@1)$ is that the distribution of single-attempt success rates can be used to predict whether power-law scaling will appear and if so, what the intercept and exponent of the power law will be. To do this, one can fit the distribution

$\hat{p}_{\mathcal{D}}(\text{pass}_i@1)$ and then *simulate* how $\text{pass}_{\mathcal{D}}@k$ will scale with $k$ (Fig. 5) using the relationship:

$$\widehat{\text{pass}_{\mathcal{D}}@k} \stackrel{\text{def}}{=}$$
$$1 - \int_0^1 (1 - \text{pass}_i@1)^k \, \hat{p}_{\mathcal{D}}(\text{pass}_i@1) \, \mathrm{d}\,\text{pass}_i@1 \,. \tag{11}$$

To empirically test this claim, we compared the standard least squares regression estimator (in log-log space) (Hoffmann et al., 2022; Caballero et al., 2022; Besiroglu et al., 2024b) against a *distributional estimator*. To motivate our distributional estimator, we first need explain a key obstacle and how the distributional estimator overcomes it. The obstacle is that there are problems or prompts whose single-attempt success probabilities $\text{pass}_i@1$ lie between $(0, 1/\text{Number of Samples})$ such that, due to finite sampling, we lack the resolution to measure. While we do not know the true single-attempt success probability for the problems that lie in this interval, we *do know how many problems fall into this left tail bucket*, and we can fit a distribution's parameters such that the distribution's probability mass in the interval $(0, 1/\text{Number of Samples})$ matches the empirical fraction of problems in this tail bucket. Thus, our distributional estimator works by first selecting a distribution (e.g., a scaled 3-parameter Beta distribution), discretizing the distribution

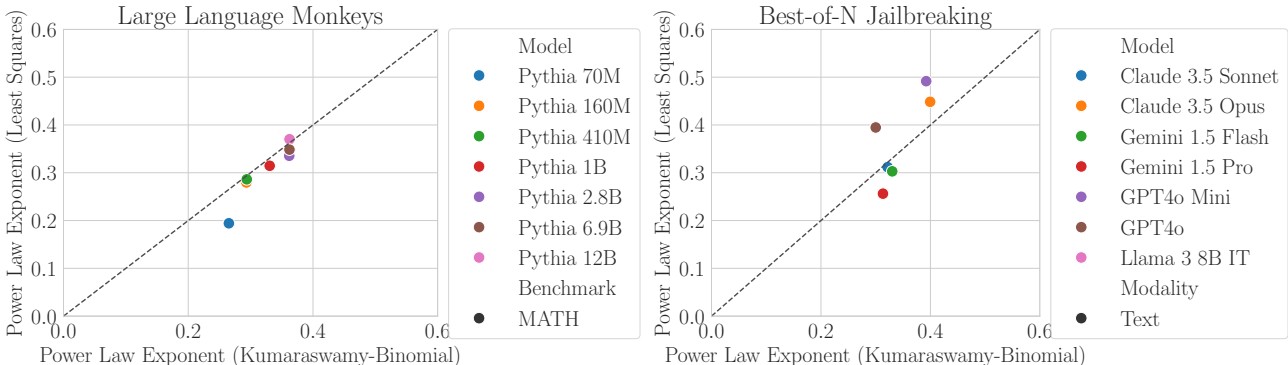

Figure 6: **Comparing Estimators of Power Law Exponents.** We compare two estimators of the power law exponent $b$ in $-\log(\text{pass}_\mathcal{D}@\text{k}) \approx ak^{-b}$ : (1) the standard least-squares estimator between $k$ and $-\log(\text{pass}_\mathcal{D}@\text{k})$ in log-log space, and (2) the distributional estimator of $\text{pass}_\text{i}@1$ assuming a scaled Kumaraswamy-Binomial distribution. Using all available data to fit both estimators, we find agreement between the least-squares estimate (ordinate) and the distribution-derived estimate (abscissa) for both Pythia models on MATH (left) and for frontier AI systems on HarmBench (right). For an explanation of why the two estimators match more closely for Large Language Monkeys than for Best-of-N Jailbreaking, see Appendix A.

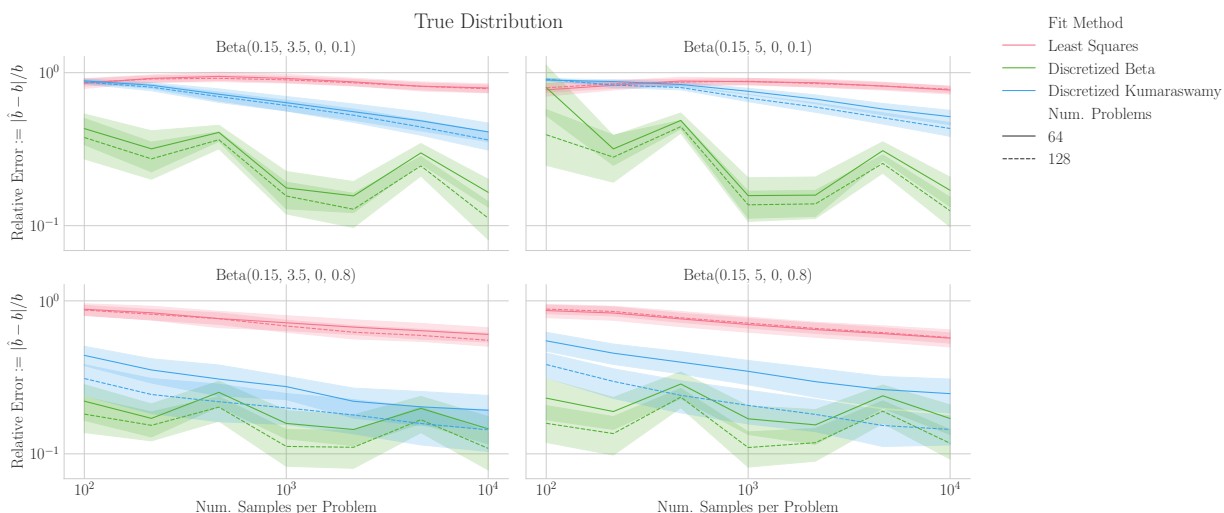

Figure 7: **Comparing Two Estimators of Power Law Exponents via Backtesting.** On synthetic data with known ground-truth power law $a\,k^{-b}$, we compare how well the least squares and the distributional estimator recover the scaling exponent $b$ as measured by the relative error $|\hat{b} - b|/b$ by backtesting: subsampling the number of problems and the number of samples per problem. We find that the distributional estimator obtains significantly better sample efficiency.

according to the sampling resolution 1/Number of Samples and performing maximum likelihood estimation under the discretized distribution's probability mass function.

We tested this distributional estimator in two different ways. First, focusing on Large Language Monkeys, we used all available real data from all problems and all samples per problem to compare the standard least squares regression estimator against the distributional estimator. We found close agreement between the two estimators (Fig. 6), giving us a sense that the two estimators yield reasonably consistent estimates under large sampling budgets.

Second, the distributional estimator also comes with another benefit: it directly provides an estimate of the power law's exponent $b$ in $a\,k^{-b}$. Estimating the power law's exponent is especially valuable because the exponent dictates how success rates are improving with increasing inference compute. To test how the distributional estimator and least squares estimator compare at recovering the true asymptotic power law exponent, we generated synthetic data so that we would have ground-truth knowledge of the true power law exponent, then backtested how the two scaling estimators compare at recovering the true exponent (Alabdulmohsin et al., 2022a; Owen, 2024) by subsampling data with fewer

problems and fewer samples per problem. We found that the distributional estimator obtains significantly better sample efficiency, with approximately an order of magnitude lower relative error $\overset{\text{def}}{=} |\hat{b} - b|/b$ compared with the least squares estimator (Fig. 7), or equivalently, $\sim 2 - 4$ orders of magnitude less inference-compute. The distributional estimator performs well even under distributional mismatch.

# 6. Related Work

Research into scaling laws of deep neural networks has a rich history spanning theoretical foundations, empirical validations, and diverse applications. The earliest investigations discovered power law scaling in simple machine learning settings (Barkai et al., 1993; Mhaskar, 1996; Pinkus, 1999). However, the modern era of scaling laws began with breakthrough studies in neural language models (Hestness et al., 2017; Kaplan et al., 2020; Brown et al., 2020b), catalyzing extensive research across multiple directions. The theoretical understanding of scaling laws has advanced significantly (Spigler et al., 2020; Bousquet et al., 2020; Hutter, 2021; Sharma & Kaplan, 2022; Maloney et al., 2022; Roberts et al., 2022; Bahri et al., 2024; Michaud et al., 2024; Paquette et al., 2024; Atanasov et al., 2024; Bordelon et al., 2024a;b; Lin et al., 2024; Brill, 2024), complemented by comprehensive empirical studies (Rosenfeld et al., 2020; Henighan et al., 2020; Gordon et al., 2021; Tay et al., 2021; Ghorbani et al., 2021; Tay et al., 2022b; Zhai et al., 2022; Alabdulmohsin et al., 2022b; Dehghani et al., 2023; Bachmann et al., 2023). In the context of language models, researchers have explored scaling behaviors in various aspects: context length (Xiong et al., 2023), in-context learning (Chan et al., 2022; Agarwal et al., 2024; Arora et al., 2024), vocabulary size (Tao et al., 2024), and jailbreaking attempts (Anil et al., 2024; Hughes et al., 2024). Studies have also investigated scaling dynamics in fine-tuning (Kalajdzievski, 2024; Zhang et al., 2024), transfer learning (Hernandez et al., 2021), and the impact of repeated data (Hernandez et al., 2022; Muennighoff et al., 2023). Architectural considerations have been extensively studied, including network design (Tay et al., 2022a; Clark et al., 2022), nested models (Kudugunta et al., 2023), pruning strategies (Rosenfeld et al., 2021), and precision requirements (Dettmers & Zettlemoyer, 2023; Kumar et al., 2024; Sun et al., 2025). Research has also addressed multimodal extensions (Aghajanyan et al., 2023; Cherti et al., 2023) and inference optimization (Sardana et al., 2023; Brown et al., 2024; Snell et al., 2024a; Wu et al., 2024; Chen et al., 2024). The field has expanded to encompass diverse domains including reinforcement learning (both single-agent (Jones, 2021; Hilton et al., 2023; Neumann & Gros, 2024) and multi-agent (Neumann & Gros, 2022)), graph networks (Liu et al., 2024), diffusion models (Mei et al., 2024; Liang et al., 2024), and associative memory models (Romani et al., 2013; Cabannes et al., 2024; Schaeffer et al., 2024c). Recent work

has explored emerging phenomena such as inverse scaling (McKenzie et al., 2024), unique functional forms (Caballero et al., 2022), scaling patterns across model families (Ruan et al., 2024; Polo et al., 2024), and downstream capabilities (Srivastava et al., 2023; Wei et al., 2022a; Hu et al., 2024; Schaeffer et al., 2024b; Snell et al., 2024b; Wu & Lo, 2024). Researchers have also investigated critical challenges including data contamination (Schaeffer, 2023; Jiang et al., 2024; Dominguez-Olmedo et al., 2024), model-data feedback loops (Dohmatob et al., 2024; Gerstgrasser et al., 2024; Kazdan et al., 2024), and overtraining effects (Gao et al., 2023; Gadre et al., 2024). Additional contributions include studies in sparse autoencoders (Gao et al., 2024), biologically-plausible backpropagation (Filipovich et al., 2022), and self-supervised learning for vision (Schaeffer et al., 2024a). Recent efforts have also focused on reconciling apparent contradictions in scaling behaviors (Besiroglu et al., 2024b; Porian et al., 2024).

# 7. Discussion and Future Directions

This work advances our mathematical understanding of how and why language model performance improves with additional inference compute through repeat sampling. By establishing rigorous theoretical foundations for these empirically-observed power laws, our work provides practitioners with principled ways to understand and predict model performance when scaling inference compute. The distributional perspective we develop explains previously puzzling deviations from power law scaling and enables more efficient estimation of scaling parameters.

Two related questions are *why* such distributional structure exists in the single-attempt success rates and whether one should expect such structure to appear in future benchmarks. We conjecture there are at least two reasons: (1) benchmark design, in that benchmarks are intentionally crafted that problems have a spread of difficulty without being too easy or too hard, and (2) selection bias, in that more interesting patterns such as power law scaling are more likely to garner more interest from the research community.

Despite focusing on scaling inference compute, our paper contributes a new hypothesis for an open question in scaling pretraining compute: *why are neural scaling laws power laws?* Just as the scaling behavior of $-\log(\text{pass}_{\mathcal{D}}@\text{k})$ only becomes clear for large $k$, so too might the scaling behavior of pretraining cross entropy with pretraining compute $C$. Specifically, suppose the pretraining cross entropy $\mathcal{L}$ as a function of pretraining compute $C$ is a sum of many functions which decay at different rates:

$$\mathcal{L}(C) = \omega\left(\frac{1}{C^\alpha}\right) + \frac{A}{C^\alpha} + o\left(\frac{1}{C^\alpha}\right),$$

where $\alpha$ is the smallest (positive) polynomial exponent and

$\omega(1/C^{\alpha})$ represents functions that decay more slowly than any polynomial. Initially, for small $C$, the dominant term may be unclear, but as pretraining compute is scaled up across $8-10$ orders of magnitude, the leading order term dominates and an approximate power law emerges:

$$\mathcal{L}(C) \approx \text{const} + \frac{A}{C^{\alpha}} + 0 \quad \text{as} \quad C \to \infty.$$

Thus, a power law relationship may only be reasonable for sufficiently large pretraining compute $C$, which in turn may require excluding the lowest pretraining compute models in order to obtain good predictions, justifying a widespread empirical practice (Kaplan et al., 2020). We designate possible functions hiding in $\omega(1/C^{\alpha})$ and $o(1/C^{\alpha})$ as *the dark matter of neural scaling laws*.

## Acknowledgments

RS acknowledges support from Stanford Data Science and the OpenAI Superalignment Fast Grant. SK acknowledges support by NSF 2046795 and 2205329, IES R305C240046, ARPA-H, the MacArthur Foundation, Schmidt Sciences, OpenAI, and Stanford HAI.

## Impact Statement

Our findings have important practical implications for the deployment of large language models, as they can help organizations more accurately forecast compute requirements and make informed trade-offs between model size, inference costs, and performance targets. The mathematical framework we develop could also generalize beyond language models to other domains where similar scaling phenomena emerge. While our work is primarily theoretical, we acknowledge that advances in language model capabilities can have broad societal impacts. We hope that better understanding these fundamental scaling behaviors will help the research community develop more efficient and reliable AI systems.

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

## A. Clarification of How Large Language Monkeys and Best-of-N Jailbreaking Sampled Data

In this manuscript, we used the phrasing of "independent attempts," which is not fully correct. In this appendix section, we clarify why we chose this terminology, what likely impacts we believe this inaccuracy may have had on our results, and how to correct the paper accordingly.

Large Language Monkeys (Brown et al., 2024) indeed drew $10,000$ independent attempts per problem, but Best-of-N Jailbreaking (Hughes et al., 2024) sampled data slightly different: for each problem, jailbreaking attempts were drawn until either a successful jailbreak was obtained or until a maximum limit of $10,000$ attempts was hit. Samples were also drawn in minibatches of size 60, making the (in)dependence of samples a bit tricky.

We omitted this nuance because it offers a second-order correction to our paper's main story while offering little additional insight. Neither of our theorems and none of our main text figures change. We suspect that this slightly different sampling procedure explains why, in Fig. 6, the estimated power law exponents between the least squares power law estimator and the distributional power law estimator deviate more significantly from identity for Best-of-N Jailbreaking than for Large Language Monkeys. A natural way to correct for this is to use a beta-negative binomial distribution rather than a beta-binomial distribution, with an additional correction for the maximum number of attempts. For more information, please see Appendix H.

## B. Estimating Success Rates Using Chen et al. (2021)'s Estimator

In this manuscript, we defined $\mathrm{pass_i@k}$ and $\mathrm{ASR_i@k}$ as:

$$\mathrm{pass_i@k} \overset{\text{def}}{=} \underset{k \text{ Attempts}}{\mathbb{E}} \left[\mathbb{I}[\text{At least 1 attempt by the model solves the } i\text{-th problem}]\right]$$

$$\mathrm{ASR_i@k} \overset{\text{def}}{=} \underset{k \text{ Attempts}}{\mathbb{E}} \left[\mathbb{I}[\text{At least 1 attempt jailbreaks the model on the } i\text{-th prompt}]\right]$$

Throughout this manuscript, to estimate $\mathrm{pass_i@k}$ and $\mathrm{ASR@k}$, we used the unbiased and lower variance estimator introduced by Chen et al. (2021): for the $i$-th problem, we sampled $n \gg k$ attempts per problem, counted the number of successful attempts $c$, and then swept $k$ to compute an estimate of $\mathrm{pass_i@k}$ for different $k$ values:

$$\widehat{\mathrm{pass_i@k}} = 1 - \frac{\binom{n-c}{k}}{\binom{n}{k}} \tag{12}$$

Two comments: Firstly, $n$ as used here has no relationship with the number of problems in the benchmark (Sec. 1), and secondly, our notation differs slightly from that of Chen et al. (2021), but the ideas are consistent. A numerically stable Python implementation of the estimator is provided in Fig. 8:

```python
def estimate_success_rate_at_k_per_problem(n: int, c: int, k: int) -> float:
    """
    :param n: number of total attempts on this problem.
    :param c: number of correct attempts on this problem.
    :param k: k in pass_i@$k$.
    """
    if n - c < k: return 1.0
    return 1.0 - np.prod(1.0 - k / np.arange(n - c + 1, n + 1))
```

Figure 8: A numerically stable unbiased estimator of $\mathrm{pass_i@k}$, introduced by Chen et al. (2021).

To reiterate a point made by Chen et al. (2021), estimating $\mathrm{pass_i@k}$ as $1 - (1 - \widehat{\mathrm{pass_i@1}})^k$ is biased (Fig. 9).

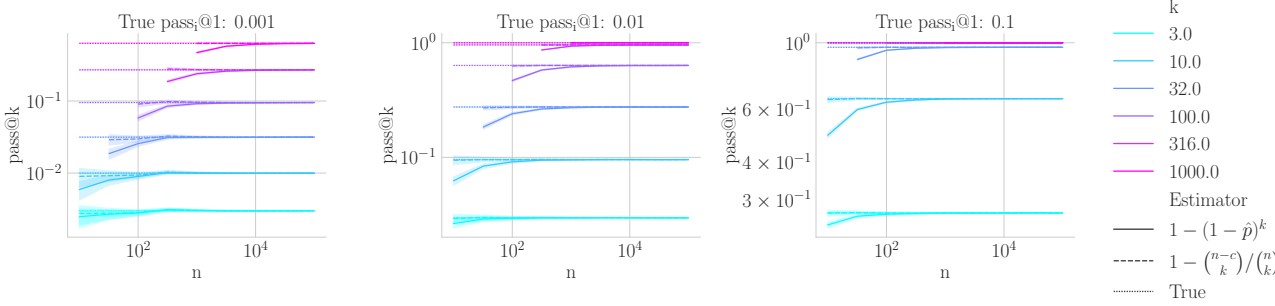

Figure 9: **Bias of Estimators of** $\mathrm{pass_i@k}$. Numerical simulations show that estimating $\mathrm{pass_i@k}$ as $1 - (1 - \widehat{\mathrm{pass_i@1}})^k$ is biased whereas the estimator of Chen et al. (2021) is not. For a mathematical proof of unbiasedness, see the original paper.

## C. Fitting Power Laws to Large Language Monkeys and Best-of-N Jailbreaking

We fit power laws to a subset of data from Large Language Monkeys (Brown et al., 2024) and from Best-of-N Jailbreaking (Hughes et al., 2024), specifically Pythia language models (Biderman et al., 2023) on the MATH benchmark (Hendrycks et al., 2021) and frontier AI models – Claude, GPT4 (OpenAI et al., 2024), Gemini (Team et al., 2024a;b) and Llama 3 (Grattafiori et al., 2024) – on the HarmBench jailbreaking benchmark (Mazeika et al., 2024). We show the functional forms and the fit parameters in Table 1 and Table 2 respectively. To fit the parameters, for Large Language Monkeys, we simply minimized the squared error between the actual and predicted $-\log(\text{pass}_{\mathcal{D}}@\text{k})$, and for Best-of-N Jailbreaking, we similarly minimized the squared error between the actual and predicted $-\log(\text{ASR}_{\mathcal{D}}@\text{k})$.

Note: Llama 3 8B IT does not exhibit power law scaling under Best-of-N Jailbreaking (shown in Fig. 1, bottom).

| Model | Benchmark | $a$ | $b$ |
|---|---|---|---|
| Pythia 70M | MATH | 8.026 | 0.194 |
| Pythia 160M | MATH | 6.591 | 0.280 |
| Pythia 410M | MATH | 5.524 | 0.286 |
| Pythia 1B | MATH | 5.452 | 0.315 |
| Pythia 2.8B | MATH | 4.104 | 0.336 |
| Pythia 6.9B | MATH | 4.255 | 0.348 |
| Pythia 12B | MATH | 4.113 | 0.370 |

Table 1: Large Language Monkeys (Brown et al., 2024) fitted power law parameters on 128 mathematical problems from MATH (Hendrycks et al., 2021).
Functional Form: $-\log(\text{pass}_{\mathcal{D}}@\text{k}) = a\,k^{-b}$.

| Model | Modality | $a$ | $b$ |
|---|---|---|---|
| Claude 3.5 Opus | Text | 2.630 | 0.448 |
| Claude 3.5 Sonnet | Text | 3.436 | 0.312 |
| GPT4o | Text | 3.639 | 0.395 |
| GPT4o Mini | Text | 3.637 | 0.492 |
| Gemini 1.5 Flash | Text | 6.158 | 0.303 |
| Gemini 1.5 Pro | Text | 6.296 | 0.256 |
| Llama 3 8B IT | Text | – | – |

Table 2: Best-of-N Jailbreaking (Hughes et al., 2024) fitted power law parameters on text jailbreak prompts from HarmBench (Mazeika et al., 2024).
Functional Form: $-\log(\text{ASR}_{\mathcal{D}}@\text{k}) = a\,k^{-b}$.
Note: Llama 3 8B Instruction Tuned (IT) does not exhibit power law scaling.

## D. Mathematical Equivalence Between Coverage and Average Success Rate

Brown et al. (2024) and Hughes et al. (2024) phrase their research in terms of "coverage", defined as the fraction of problems that can be solved or the fraction of prompts that can jailbreak a model, but as Brown et al. (2024) comment and we here derive, the coverage is mathematically equivalent to the average $\text{pass}_i@k$ (equivalently, $\text{ASR}@k$. due to two simple probabilistic primitives: (1) linearity of expectation, (2) the expectation of an indictor random variable of some event is the probability of said event and (3) the definition of $\text{pass}_i@k$:

$$
\begin{aligned}
\mathbb{E}_{\substack{\text{Prompts} \\ \text{Attempts}}}\Big[\text{Coverage}\Big] &\stackrel{\text{def}}{=} \mathbb{E}_{\substack{\text{Problems} \\ \text{Attempts}}}\Big[\text{Fraction of Problems Solved After } k \text{ Attempts}\Big] \\
&= \mathbb{E}_{\text{Problems}}\Big[\mathbb{E}_{\text{Attempts}|\text{Problem}}\big[\mathbb{I}\big[\text{Problem Solved After } k \text{ Attempts}\big]\big]\Big] \\
&= \mathbb{E}_{\text{Problems}}\Big[\text{pass}_{\text{problem}}@k\Big] \\
&= \text{pass}_{\mathcal{D}}@k
\end{aligned}
$$

In our work, we prefer phrasing along the lines of "success rate" over "coverage" because success rate avoids coverage's binary implication that each problem/prompt is either "solved" or "not solved".

## E. Aggregate Power Laws from a Probability Distribution over Exponential Functions

### E.1. Preliminaries: Power Laws from Weighted Exponential Functions

A known result is that power laws can emerge from appropriately weighted sums of exponential functions, e.g., (Bochud & Challet, 2006; Elkies, 2016; Bousquet et al., 2020). For a concrete example with a short proof:

$$x^{-r} = \frac{1}{\Gamma(r)} \int_0^\infty p^{r-1} e^{-px} \, dp, \tag{13}$$

where $\Gamma(r) \stackrel{\text{def}}{=} \int_0^\infty s^{r-1} e^{-s} \, ds$ is the Gamma function. The proof is via u-substitution $u \stackrel{\text{def}}{=} p\,x$:

$$\frac{1}{\Gamma(r)} \int_0^\infty p^{r-1} e^{-px} \, dp = \frac{1}{\Gamma(r)} \int_0^\infty (u/x)^{r-1} e^{-u} \, \frac{du}{x} \tag{14}$$

$$= \frac{1}{\Gamma(r)} x^{-r} \int_0^\infty u^{r-1} e^{-u} \, du \tag{15}$$

$$= \frac{1}{\Gamma(r)} x^{-r} \, \Gamma(r) \tag{16}$$

$$= x^{-r} \tag{17}$$

In our particular context, we are interested in the scaling with $k$ of the expected success rate over problems sampled from the benchmark's data distribution:

$$\text{pass}_\mathcal{D}@\text{k} \stackrel{\text{def}}{=} \mathop{\mathbb{E}}_{\text{pass}_i@1 \sim \mathcal{D}} \left[ \text{pass}_i@\text{k} \right] \tag{18}$$

distribution (over problems in a benchmark) of $\text{pass}_i@\text{k}$ scores that yields power law scaling with respect to the number of attempts $k$:

$$-\log \left( \frac{1}{n} \sum_{i=1}^n \text{pass}_i@\text{k} \right) \approx a k^{-b}. \tag{19}$$

for constants $a, b > 0$.

### E.2. Delta Distribution: $\text{pass}_i@1 \sim \delta(p), p \in (0, 1)$

To start with a negative result, we will show that not all distributions of the per-problem success probabilities $\text{pass}_i@1$ yield aggregate power law scaling. Suppose that the model's $\text{pass}_i@1$ probabilities across the benchmarks' problems are all exactly $p \in (0, 1)$. For brevity, let $p_i \stackrel{\text{def}}{=} \text{pass}_i@1$. Then the aggregate success rate is:

$$\mathbb{E}_{p_i \sim \delta(p)}[\text{pass}_i@\text{k}] = 1 - \mathbb{E}_{p_i}[(1 - p_i)^k] \tag{20}$$

$$= \int_0^1 \delta(p) \, (1 - p_i)^k \, dp_i \tag{21}$$

$$= (1 - p)^k. \tag{22}$$

Recalling that the expansion of $\log(\cdot)$ for small $x$ is $-\log(1 - x) = x + O(x^2)$, in our case, we obtain:

$$-\log \left( 1 - \mathbb{E}_{p_i \sim \delta(p)}[\text{pass}@\text{k}] \right) = (1 - p)^k + O((1 - p)^{2k}) = (1 - p)^k + o((1 - p)^k). \tag{23}$$

Thus, in the large $k$ regime, we find the negative log aggregate success rate exhibits *exponential* scaling with $k$ as we intuitively expect.

### E.3. Uniform Distribution: $\text{pass}_i@1 \sim \text{Uniform}(\alpha, \beta)$

Suppose $\text{pass}_i@1$ probabilities follow a uniform distribution $\text{Uniform}(\alpha, \beta)$ where $0 \leq \alpha < \beta \leq 1$. The aggregate success rate after $k$ attempts is defined as:

$$\text{pass}_{\text{Uniform}(\alpha, \beta)}@k \stackrel{\text{def}}{=} 1 - \mathbb{E}\big[(1 - p)^k\big].$$

If $p \sim \text{Uniform}(\alpha, \beta)$, the expectation of $(1 - p)^k$ is:

$$\mathbb{E}\big[(1 - p)^k\big] \;=\; \frac{1}{\beta - \alpha} \int_\alpha^\beta (1 - p)^k \, dp.$$

Evaluating the integral gives:

$$\mathbb{E}\big[(1 - p)^k\big] \;=\; \frac{(1 - \alpha)^{k+1} - (1 - \beta)^{k+1}}{(\beta - \alpha) \cdot (k + 1)}.$$

Thus, the aggregate success rate becomes:

$$\text{pass}_{\text{Uniform}(\alpha,\beta)} @k \;=\; 1 \;-\; \frac{(1 - \alpha)^{k+1} - (1 - \beta)^{k+1}}{(\beta - \alpha) \cdot (k + 1)}.$$

**Case A:** $\alpha > 0$  If $\alpha > 0$, then both $(1 - \alpha)$ and $(1 - \beta)$ are strictly less than 1. As $k \to \infty$, $(1 - \alpha)^{k+1}$ and $(1 - \beta)^{k+1}$ decay exponentially. Hence:

$$\mathbb{E}\big[(1 - p)^k\big] \;\sim\; \frac{(1 - \alpha)^{k+1}}{(\beta - \alpha) \cdot (k + 1)},$$

and $\text{pass}_{\text{Uniform}(\alpha,\beta)} @k$ approaches 1 exponentially fast:

$$\text{pass}_{\text{Uniform}(\alpha,\beta)} @k \;\sim\; 1 - \frac{(1 - \alpha)^{k+1}}{(\beta - \alpha) \cdot (k + 1)}.$$

Thus, the negative log of the aggregate success rate decays exponentially:

$$-\log\big(\text{pass}_{\text{Uniform}(\alpha,\beta)} @k\big) \;\sim\; e^{-\Omega(k)}.$$

**Case B:** $\alpha = 0$  When $\alpha = 0$, the uniform distribution is over $[0, \beta]$. In this case:

$$\mathbb{E}\big[(1 - p)^k\big] \;=\; \frac{1}{\beta} \cdot \frac{1 - (1 - \beta)^{k+1}}{k + 1}.$$

For large $k$, $(1 - \beta)^{k+1}$ becomes exponentially small, and:

$$\mathbb{E}\big[(1 - p)^k\big] \;\sim\; \frac{1}{\beta} \cdot \frac{1}{k + 1}.$$

The aggregate success rate is then:

$$\text{pass}_{\text{Uniform}(0,\beta)} @k \;\sim\; 1 - \frac{1}{\beta \cdot k}.$$

The negative log exhibits power-law scaling:

$$-\log\big(\text{pass}_{\text{Uniform}(0,\beta)} @k\big) \;\sim\; \frac{1}{\beta} \cdot \frac{1}{k}.$$

**Special Case:** $\text{Uniform}(0, 1)$  If $\beta = 1$, the distribution is uniform on $[0, 1]$. In this case:

$$\mathbb{E}\big[(1 - p)^k\big] \;=\; \frac{1}{k + 1},$$

and the success rate becomes:

$$\text{pass}_{\text{Uniform}(0,1)} @k \;=\; 1 - \frac{1}{k + 1}.$$

For large $k$:

$$-\log\big(\text{pass}_{\text{Uniform}(0,1)} @k\big) \;\sim\; \frac{1}{k}.$$

**E.4. 2-Parameter Beta Distribution:** $\text{pass}_i@1 \sim \text{Beta}(\alpha, \beta)$

Suppose that the model's $\text{pass}_i@1$ probabilities across the benchmark problems follow a Beta distribution:

$$\text{pass}_i@1 \sim \text{Beta}(\alpha, \beta)$$

The probability density function of this distribution over the support $x \in (0, 1)$ is:

$$f(x; \alpha, \beta) \stackrel{\text{def}}{=} \frac{1}{B(\alpha, \beta)} x^{\alpha-1} (1-x)^{\beta-1}, \tag{24}$$

where $\alpha > 0, \beta > 0$ and $B(\cdot, \cdot)$ is the Beta function. For brevity, let $p_i \stackrel{\text{def}}{=} \text{pass}_i@1$. Under our assumed Beta distribution:

$$\text{pass}_{\text{Beta}(\alpha,\beta)}@k \stackrel{\text{def}}{=} 1 - \mathbb{E}_{p_i \sim \text{Beta}(\alpha,\beta)}[(1 - p_i)^k] \tag{25}$$

$$= 1 - \int_0^1 \frac{p_i^{\alpha-1}(1 - p_i)^{\beta-1}}{B(\alpha, \beta)} (1 - p_i)^k \, dp_i \tag{26}$$

$$= 1 - \frac{\Gamma(\alpha + \beta)}{\Gamma(\alpha)\Gamma(\beta)} \frac{\Gamma(\alpha)\Gamma(\beta + k)}{\Gamma(\alpha + \beta + k)} \tag{27}$$

where $\Gamma(\cdot)$ is again the Gamma function. The $\Gamma(\alpha)$ terms cancel, and a standard asymptotic result of the gamma function for large $k$ tells us that:

$$\frac{\Gamma(\beta + k)}{\Gamma(\alpha + \beta + k)} \sim k^{-\alpha}, \tag{28}$$

and thus:

$$\frac{\Gamma(\alpha + \beta)}{\Gamma(\beta)} \frac{\Gamma(\beta + k)}{\Gamma(\alpha + \beta + k)} \sim \frac{\Gamma(\alpha + \beta)}{\Gamma(\beta)} k^{-\alpha}. \tag{29}$$

Recalling again that the expansion of $\log(\cdot)$ for small $x$ is $-\log(1 - x) = x + O(x^2)$, in our case, we obtain:

$$-\log\left(\text{pass}_{\mathcal{D}}@k\right) = \frac{\Gamma(\alpha + \beta)}{\Gamma(\beta)} k^{-\alpha} + O(k^{-2\alpha}) = \frac{\Gamma(\alpha + \beta)}{\Gamma(\beta)} k^{-\alpha} + o(k^{-\alpha}). \tag{30}$$

From this final result, we see that under a Beta distribution and in the large $k$ regime, the negative log aggregate success rate exhibits polynomial (power-law) scaling with $k$ for exponent $\alpha$

**E.5. Kumaraswamy Distribution:** $\text{pass}_i@1 \sim \text{Kumaraswamy}(\alpha, \beta)$

Next, suppose the model's $\text{pass}_i@1$ probabilities follow a Kumaraswamy distribution. The probability density function of this distribution over the support $x \in (0, 1)$ is:

$$f(x; \alpha, \beta) \stackrel{\text{def}}{=} \alpha \beta \, x^{\alpha-1} (1 - x^\alpha)^{\beta-1} \tag{31}$$

Again for brevity, let $p_i \stackrel{\text{def}}{=} \text{pass}_i@1$. Under our assumed Kumaraswamy distribution:

$$\text{pass}_{\text{Kumaraswamy}(\alpha,\beta)}@k \stackrel{\text{def}}{=} 1 - \mathbb{E}_{p_i \sim \text{Kumaraswamy}(\alpha,\beta)}[(1 - p_i)^k] \tag{32}$$

$$= 1 - \int_0^1 (1 - p)^k \cdot \alpha \beta \, p^{\alpha-1} (1 - p^\alpha)^{\beta-1} dp. \tag{33}$$

Define the integral

$$I_k \stackrel{\text{def}}{=} \mathbb{E}\left((1 - p)^k\right) = \int_0^1 (1 - x)^k \alpha \beta \, x^{\alpha-1} \left(1 - x^\alpha\right)^{\beta-1} \, dx. \tag{34}$$

We aim to analyze $I_k$ for large $k$. Notice that $(1 - x)^k$ is exponentially small in $k$ unless $x$ is very close to 0. Thus, intuitively, most of the contribution to $I_k$ arises from $x \in [0, O(1/k)]$.

**Step 1: Split the integral into two parts.** Fix a constant $c > 0$. Write

$$I_k = \int_0^{c/k} [\cdots] \, dx + \int_{c/k}^1 [\cdots] \, dx \overset{\text{def}}{=} I_{k,\text{left}} + I_{k,\text{right}},$$

where $[\cdots]$ indicates the same integrand. In the region $x \in [c/k, 1]$, we have $(1 - x)^k \leq e^{-kx} \leq e^{-c}$. Hence $I_{k,\text{right}} = O(e^{-c})$. Since $c$ can be made arbitrarily large, $I_{k,\text{right}}$ becomes negligible compared to any polynomial in $1/k$.

**Step 2: Approximate the integrand in the small-$x$ region.** On $[0, c/k]$, we use the approximation $\log(1 - x) = -x + O(x^2)$. Thus

$$(1 - x)^k = \exp\left(k \log(1 - x)\right) = \exp\left(-kx + O(kx^2)\right).$$

Since $x \leq c/k$ implies $kx^2 \leq c^2/k = O(1/k)$, and $\exp(\epsilon) = 1 + O(\epsilon)$, we get

$$(1 - x)^k = \exp(-kx) \exp(O(1/k)) = \exp(-kx)\left(1 + O\left(\tfrac{1}{k}\right)\right).$$

Furthermore, since $(1 - y)^m = 1 - my + O(y^2)$, for small $x$

$$(1 - x^\alpha)^{\beta - 1} = 1 - (\beta - 1)x^\alpha + O(x^{2\alpha}) = 1 + O(x^\alpha).$$

In the region $x \leq c/k$, that error is $O(k^{-\alpha})$. Hence, within the small-$x$ region, the integrand

$$(1 - x)^k \, \alpha \beta \, x^{\alpha - 1} \left(1 - x^\alpha\right)^{\beta - 1}$$

can be approximated by

$$\alpha \beta \, x^{\alpha - 1} \, e^{-kx} + O\left(k^{-\alpha} x^{\alpha - 1} e^{-kx}\right).$$

Thus

$$I_{k,\text{left}} = \int_0^{c/k} \alpha \beta \, x^{\alpha - 1} e^{-kx} \, dx + O\left(k^{-\alpha} \int_0^{c/k} x^{\alpha - 1} e^{-kx} \, dx\right) + O(e^{-c}).$$

**Step 3: Substitution $u \overset{\text{def}}{=} kx$.** To handle $\int_0^{c/k} x^{\alpha - 1} e^{-kx} \, dx$, we substitute $u = kx$. Then $x = u/k$, $dx = du/k$, and the upper limit $x = c/k$ becomes $u = c$. Hence,

$$\int_0^{c/k} x^{\alpha - 1} e^{-kx} \, dx = \int_0^c \left(\tfrac{u}{k}\right)^{\alpha - 1} e^{-u} \, \frac{du}{k}$$

$$= k^{-\alpha} \int_0^c u^{\alpha - 1} e^{-u} \, du.$$

As $c \to \infty$, $\int_0^c u^{\alpha - 1} e^{-u} \, du \to \Gamma(\alpha)$, and for finite $c$ the remainder is $O(e^{-c})$. Therefore,

$$\int_0^1 x^{\alpha - 1} e^{-kx} \, dx = k^{-\alpha} \Gamma(\alpha) + O\left(k^{-\alpha} e^{-c}\right),$$

and absorbing the constant $c$ into big-$O$ notation gives

$$\int_0^1 x^{\alpha - 1} e^{-kx} \, dx = k^{-\alpha} \Gamma(\alpha) + O\left(k^{-\alpha - \epsilon}\right) \quad \text{for some } \epsilon > 0.$$

Multiplying by the factor $\alpha \beta$, we deduce that

$$I_k = \alpha \beta \Gamma(\alpha) k^{-\alpha} + O\left(k^{-\alpha - \epsilon}\right).$$

**Step 4: Final conclusion for the success rate.** Recall $\text{pass}_{\text{Kumaraswamy}(\alpha,\beta)}@\text{k} = 1 - I_k$. Hence

$$\text{pass}_{\text{Kumaraswamy}(\alpha,\beta)}@\text{k} = 1 - \alpha\beta\,\Gamma(\alpha)\,k^{-\alpha} + O(k^{-\alpha-\epsilon}).$$

Since this tends to 1, its negative log is governed by the magnitude of $\alpha\beta\,\Gamma(\alpha)\,k^{-\alpha}$. Using the expansion $-\log(1-y) = y + O(y^2)$ as $y \to 0$, we get

$$-\log\left(\text{pass}_{\text{Kumaraswamy}(\alpha,\beta)}@\text{k}\right) = \alpha\beta\,\Gamma(\alpha)\,k^{-\alpha} + o(k^{-\alpha}).$$

That is precisely polynomial (power-law) decay in the negative log success rate with exponent $\alpha$.

**E.6. Continuous Bernoulli Distribution:** $\text{pass}_i@1 \sim \text{ContinousBernoulli}(\lambda)$

Next, suppose the model's $\text{pass}_i@1$ probabilities follow a Continuous Bernoulli distribution. The probability density function of this distribution over the support $x \in [0,1]$ is:

$$f(x;\lambda) \overset{\text{def}}{=} C(\lambda)\lambda^x(1-\lambda)^{1-x} \tag{35}$$

$$C(\lambda) \overset{\text{def}}{=} \begin{cases} 2 & \text{if } \lambda = 1/2 \\ \frac{2\tanh^{-1}(1-2\lambda)}{1-2\lambda} & \text{otherwise} \end{cases}. \tag{36}$$

The density can equivalently be rewritten in a more convenient form for our purposes:

$$f(x;\lambda) = C(\lambda)\lambda^x(1-\lambda)(1-\lambda)^{-x} = C(\lambda)(1-\lambda)\left(\frac{\lambda}{1-\lambda}\right)^x \tag{37}$$

Because the individual success probability is low in our data, we shall consider the small $\lambda < 1/2$ regime. We follow the same approach as with the Kumaraswamy distribution.

**Step 1: Write the aggregate pass rate.** The aggregate pass rate is defined as:

$$\text{pass}_{\text{ContinuousBernoulli}(\lambda)}@\text{k} = 1 - I_k, \quad \text{where} \quad I_k \overset{\text{def}}{=} \int_0^1 (1-p)^k f(p;\lambda)\,dp.$$

Substituting the density $f(p;\lambda)$, we get:

$$I_k = \int_0^1 (1-p)^k C(\lambda)\,\lambda^p\,(1-\lambda)^{1-p}\,dp.$$

**Step 2: Simplify using an exponential form.** Using the exponential rewriting:

$$\lambda^p(1-\lambda)^{1-p} = (1-\lambda)\exp\left(p\log\left(\tfrac{\lambda}{1-\lambda}\right)\right),$$

the integral becomes:

$$I_k = C(\lambda)(1-\lambda)\int_0^1 (1-p)^k \exp\left(p\log\left(\tfrac{\lambda}{1-\lambda}\right)\right)dp.$$

**Step 3: Dominance of the small-$p$ region.** For large $k$, $(1-p)^k$ decays exponentially unless $p$ is close to 0. Thus, the main contribution to the integral arises from the region $p \in [0, c/k]$, where $c > 0$ is a constant. Decompose the integral:

$$I_k = \int_0^{c/k}[\cdots]\,dp + \int_{c/k}^1 [\cdots]\,dp \overset{\text{def}}{=} I_{k,\text{left}} + I_{k,\text{right}}.$$

In the region $p \in [c/k, 1]$, we have $(1-p)^k \le e^{-kp} \le e^{-c}$, making $I_{k,\text{right}} = O(e^{-c})$, which is negligible compared to $1/k$. Thus, we focus on $I_{k,\text{left}}$:

$$I_{k,\text{left}} = C(\lambda)(1-\lambda)\int_0^{c/k}(1-p)^k \exp\left(p\log\left(\tfrac{\lambda}{1-\lambda}\right)\right)dp.$$

**Step 4: Approximate the integrand.** For $p \in [0, c/k]$, use the same approximations from the Kumaraswamy derivation:

$$(1-p)^k = e^{-kp}\left(1 + O(p)\right), \quad \exp\left(p\log\left(\frac{\lambda}{1-\lambda}\right)\right) = 1 + O(p).$$

Thus, the integrand becomes:

$$(1-p)^k \exp\left(p\log\left(\frac{\lambda}{1-\lambda}\right)\right) = e^{-kp}\left(1 + O(p)\right).$$

**Step 5: Change of variables.** Let $u \stackrel{\text{def}}{=} kp$, so $p = u/k$ and $dp = du/k$. The integral becomes:

$$I_{k,\text{left}} = C(\lambda)\,(1-\lambda)\int_0^c e^{-u}\left(1 + O(u/k)\right)\frac{du}{k}.$$

Split the integral:

$$I_{k,\text{left}} = \frac{C(\lambda)\,(1-\lambda)}{k}\int_0^c e^{-u}\,du + O\left(\frac{1}{k^2}\right).$$

As $c \to \infty$, $\int_0^c e^{-u}\,du \to 1$. Thus:

$$I_{k,\text{left}} = \frac{C(\lambda)\,(1-\lambda)}{k} + O\left(\frac{1}{k^2}\right).$$

Since $I_{k,\text{right}} = O(e^{-c})$ is negligible, we have:

$$I_k = \frac{C(\lambda)\,(1-\lambda)}{k} + O\left(\frac{1}{k^2}\right).$$

**Step 7: Final conclusion for the success rate.** Recall:

$$\text{pass}_{\text{ContinuousBernoulli}(\lambda)}@k = 1 - I_k.$$

For large $k$, this implies:

$$\text{pass}_{\text{ContinuousBernoulli}(\lambda)}@k = 1 - \frac{C(\lambda)\,(1-\lambda)}{k} + O\left(\frac{1}{k^2}\right).$$

Using the expansion $-\log(1-y) = y + O(y^2)$ for small $y$, we find:

$$-\log\left(\text{pass}_{\text{ContinuousBernoulli}(\lambda)}@k\right) = C(\lambda)\,(1-\lambda)k^{-1} + o(k^{-1}).$$

That is precisely polynomial (power-law) decay in the negative log success rate with exponent $-1$.

As a side comment, recall that $\tanh^{-1}(x) = \frac{1}{2}\log\left(\frac{1+x}{1-x}\right)$, the normalizing constant $C(\lambda)$ can be rewritten as:

$$C(\lambda) = \frac{2}{1-2\lambda}\frac{1}{2}\log\left(\frac{1+(1-2\lambda)}{1-(1-2\lambda)}\right) = \frac{1}{1-2\lambda}\log\left(\frac{1-\lambda}{\lambda}\right). \tag{38}$$

Thus, for small $\lambda$, note that $C(\lambda) \approx \log(1/\lambda) = -\log(\lambda)$. For $k \ll -\log(\lambda)$, the $1/k$ formula is valid. However, near $k \approx -\log(\lambda)$, the leading term $-\log(\lambda)/k$ becomes of order 1, and for $k \gg -\log(\lambda)$, the success rate is now very close to 1. Consequently, we see that if $\lambda$ is very small, there is a soft cutoff scale around $k \approx -\log(\lambda)$.

### E.7. Any Continuous Distribution with $p(\text{pass}_i@1) = c > 0$

Suppose that the distribution over $\text{pass}_i@1$ is continuous and has constant non-zero density near 0:

$$f(0) = c > 0 \tag{39}$$

Because the density is continuous at 0 with $f(0) = c > 0$, there exist some $\delta > 0$ such that:

$$f(p) = c + O(p) \qquad \text{for all } p \in [0, \delta]. \tag{40}$$

Because the small $\text{pass}_i@1$ region dominates for large $k$, a similar argument to the Kumaraswamy argument and Continuous Bernoulli argument yields power law scaling with respect to $k$ with exponent $-1$:

$$-\log\left(\text{pass}_{\mathcal{D}}@k\right) = c\,k^{-1} + o(k^{-1}). \tag{41}$$

This result is consistent with the Continuous Bernoulli, where $c$ is given by $f_{\text{ContinuousBernoulli}(\lambda)}(0; \lambda) = C(\lambda)(1-\lambda)$ for $\lambda < 1/2$. This result reveals that the Continous Bernoulli is just one instance of a larger family: any continuous distribution with non-zero constant density at $\text{pass}_i@1 = 0$ will exhibit power law scaling with exponent $-1$.

### E.8. Reciprocal Distribution: $\text{pass}_i@1 \sim \text{Reciprocal}(a, b)$

Next, suppose the model's $\text{pass}_i@1 \sim \text{Reciprocal}(a, b)$ distribution with $0 < a < b < 1$. The probability density function of this distribution over the support $x \in [a, b]$ is:

$$f(x; a, b) = \frac{1}{(\log(b) - \log(a))\, x} \tag{42}$$

As with the other distributions, the aggregate success rate after $k$ attempts is:

$$\text{pass}_{\text{Reciprocal}(a,b)}@k = \mathbb{E}\left[\text{pass}_i@k\right] = 1 - I_k, \quad \text{where} \quad I_k \overset{\text{def}}{=} \int_{x=a}^{b} (1-x)^k \frac{1}{(\log b - \log a)\, x}\, dx.$$

We aim to show that $I_k$ is on the order of $\frac{(1-a)^k}{k}$. The main contribution to the integral arises from the vicinity of $x = a$, because $(1-x)^k$ decays rapidly as $x$ grows away from $a$.

**Step 1: Change of variable.** Define $y \overset{\text{def}}{=} x - a$, so the domain $x \in [a, b]$ becomes $y \in [0, b - a]$. Then

$$(1-x)^k = \left((1-a) - y\right)^k,$$

and

$$I_k = \frac{1}{\log(b/a)} \int_{y=0}^{b-a} \left((1-a) - y\right)^k \frac{1}{a + y}\, dy.$$

**Step 2: Expansion near $y = 0$.** For small $y$, write $(1-a) - y = (1-a)\left(1 - \frac{y}{1-a}\right)$; hence

$$\log\left((1-a) - y\right) = \log(1-a) + \log\left(1 - \frac{y}{1-a}\right).$$

Using $\log(1 - z) = -z + O(z^2)$ for small $z$, we get

$$\log\left((1-a) - y\right) = \log(1-a) - \frac{y}{1-a} + O\left(\frac{y^2}{(1-a)^2}\right),$$

so

$$(1 - a - y)^k = \exp\left(k\log(1-a) - k\frac{y}{1-a} + O\left(\frac{k\,y^2}{(1-a)^2}\right)\right).$$

In particular, for $y$ up to $c/k$, the term $k\,y^2 = O(1)$ remains bounded, so

$$(1 - a - y)^k = (1-a)^k \exp\left(-\frac{k\,y}{1-a}\right)\left[1 + O\left(\tfrac{1}{k}\right)\right].$$

**Step 3: The integral is dominated by $y \in [0, O(\tfrac{1}{k})]$.** For large $k$, $\exp\left(-\frac{k\,y}{1-a}\right)$ decays quickly once $y$ exceeds a multiple of $\frac{1-a}{k}$. Consequently, the integral from $y = c_0/k$ to $b - a$ is exponentially small in $k$. On $[0, c_0/k]$, we also have $(a + y)^{-1} = \frac{1}{a} + O\left(\tfrac{1}{k}\right)$. Thus

$$I_k = \frac{1}{\log(b/a)} \int_{y=0}^{c_0/k} (1 - a - y)^k \frac{1}{a + y}\, dy + \text{(exponentially small tail)}.$$

Substitute our approximation from Step 2 into the integrand:

$$(1 - a - y)^k \, \frac{1}{a + y} \;=\; (1 - a)^k \, \exp\!\left(-\tfrac{k\,y}{1-a}\right) \left[\tfrac{1}{a} + O\!\left(\tfrac{1}{k}\right)\right].$$

**Step 4: Change variable** $u = \frac{k\,y}{1-a}$. Then $y = \frac{(1-a)\,u}{k}$ and $\mathrm{d}y = \frac{1-a}{k}\,\mathrm{d}u$. The upper limit $y = c_0/k$ corresponds to $u = c_0\left(\frac{1-a}{1}\right)$, so

$$\int_{y=0}^{c_0/k} \exp\!\left(-\tfrac{k\,y}{1-a}\right) \mathrm{d}y \;=\; \int_{u=0}^{c_0\,(1-a)} e^{-u}\, \frac{1 - a}{k}\,\mathrm{d}u.$$

Letting $c_0 \to \infty$ only contributes an $e^{-c_0(1-a)}$ factor to the tail, which vanishes. Hence

$$\int_{y=0}^{\infty} \exp\!\left(-\tfrac{k\,y}{1-a}\right) \mathrm{d}y \;=\; \frac{1 - a}{k} \int_{u=0}^{\infty} e^{-u}\,\mathrm{d}u \;=\; \frac{1 - a}{k}.$$

Putting all factors together,

$$I_k \;=\; \frac{1}{\log(b/a)}\, (1 - a)^k \left[\tfrac{1}{a} + O\!\left(\tfrac{1}{k}\right)\right] \frac{1 - a}{k} \;+\; \text{(exponentially small in } k).$$

Thus in big-Theta form,

$$I_k \;=\; \Theta\!\left(\tfrac{(1-a)^k}{k}\right).$$

**Conclusion.** Since $\mathrm{pass}_{\mathrm{Reciprocal(a,b)}}@\mathrm{k} = 1 - I_k$, we get

$$\mathrm{pass}_{\mathrm{Reciprocal(a,b)}}@\mathrm{k} \;=\; 1 \;-\; \Theta\!\left(\tfrac{(1-a)^k}{k}\right).$$

Moreover, using $-\log(1 - y) = y + O(y^2)$ for small $y$, it follows that

$$-\log\!\Big(\mathrm{pass}_{\mathrm{Reciprocal(a,b)}}@\mathrm{k}\Big) \;=\; \Theta\!\left(\tfrac{(1-a)^k}{k}\right).$$

Hence the negative log aggregate success rate converges to 1 *exponentially fast* in $k$, which is *not* a power law in $k$.

**Sufficient Condition for Power-Law Scaling in Negative Log of Aggregate Success**

**Theorem E.1.** *Let $\mathcal{D}$ be a probability distribution on $[0,1]$ with PDF $f(p)$. Suppose there exist constants $b > 0$, $C > 0$, $\theta > 0$ and $\delta > 0$ such that, for all $0 < p < \delta$, we have*

$$f(p) \; = \; C\,p^{b-1} \; + \; O\big(p^{b-1+\theta}\big).$$

*Then, for large $k$,*

$$1 \; - \; \mathrm{pass}_{\mathcal{D}}@\mathrm{k} \; = \; C\,\Gamma(b)\,k^{-b} \; + \; O\big(k^{-b-\min(1,\theta)}\big),$$

*which implies*

$$-\log\big(\mathrm{pass}_{\mathcal{D}}@\mathrm{k}\big) \; = \; C\,\Gamma(b)\,k^{-b} \; + \; o\big(k^{-b}\big).$$

*Equivalently, including the leading constant),*

$$-\log\big(\mathrm{pass}_{\mathcal{D}}@\mathrm{k}\big) \; \sim \; C\,\Gamma(b)\,k^{-b}.$$

*Proof.* **Step 1. Decompose the key integral.**
Define

$$I_k \; \overset{\text{def}}{=} \; 1 \; - \; \mathrm{pass}_{\mathcal{D}}@\mathrm{k} \; = \; \int_0^1 (1-p)^k\, f(p)\,\mathrm{d}p.$$

For a positive constant $c > 0$, split $I_k$:

$$I_k \; = \; \int_0^{c/k} (1-p)^k\, f(p)\,\mathrm{d}p \; + \; \int_{c/k}^1 (1-p)^k\, f(p)\,\mathrm{d}p \; \overset{\text{def}}{=} \; I_{k,\text{left}} \; + \; I_{k,\text{right}}.$$

**Right Tail Bound ($I_{k,\text{right}}$).**   For $p \geq c/k$, observe $(1-p)^k \leq e^{-kp} \leq e^{-c}$. Hence

$$I_{k,\text{right}} \; = \; \int_{c/k}^1 (1-p)^k\, f(p)\,\mathrm{d}p \; \leq \; e^{-c} \int_0^1 f(p)\,\mathrm{d}p \; = \; e^{-c}.$$

Since $c$ can be made arbitrarily large, $e^{-c}$ can be driven below *any* power of $1/k$. Thus $I_{k,\text{right}} = o\big(k^{-\alpha}\big)$ for any $\alpha > 0$. We may therefore focus on

$$I_{k,\text{left}} \; = \; \int_0^{c/k} (1-p)^k\, f(p)\,\mathrm{d}p,$$

knowing that $I_{k,\text{right}}$ is negligible in polynomial-type estimates.

**Step 2. Use the assumed behavior of $f(p)$ near $p = 0$.**
By hypothesis, for $p$ up to some $\delta > 0$,

$$f(p) \; = \; C\,p^{b-1} + O\big(p^{b-1+\theta}\big).$$

Choose $c/k < \delta$, so $p \leq c/k < \delta$ for $p$ in the left integral. Then

$$I_{k,\text{left}} \; = \; \int_0^{c/k} (1-p)^k \Big[ C\,p^{b-1} + O\big(p^{b-1+\theta}\big) \Big]\,\mathrm{d}p.$$

Split it into main term and error term:

$$I_{k,\text{left}} \; = \; C \int_0^{c/k} (1-p)^k\, p^{b-1}\,\mathrm{d}p \; + \; \int_0^{c/k} (1-p)^k\, O\big(p^{b-1+\theta}\big)\,\mathrm{d}p.$$

Denote these $T_{\text{main}}$ and $T_{\text{err}}$, respectively.

**Step 3. Approximate $(1-p)^k$ by $e^{-kp}$ and control the error.**
For $p$ in $[0, c/k]$, expand $\log(1-p) = -p + O(p^2)$. Thus

$$(1-p)^k \; = \; \exp\big(k \log(1-p)\big) \; = \; e^{-kp}\, \exp\big(O(kp^2)\big) \; = \; e^{-kp}\big[1 + O(kp^2)\big].$$

Since $p \leq c/k$, we get $k p^2 \leq c^2/k$, which is bounded for large $k$. Consequently,

$$(1 - p)^k = e^{-kp} + O(k p^2 e^{-kp}).$$

We will use this in both $T_{\text{main}}$ and $T_{\text{err}}$.

**Step 4. Main term $T_{\text{main}}$.**

$$T_{\text{main}} = C \int_0^{c/k} (1 - p)^k p^{b-1} \, dp.$$

Substituting $(1 - p)^k = e^{-kp} + O(k p^2 e^{-kp})$,

$$T_{\text{main}} = C \int_0^{c/k} e^{-kp} p^{b-1} \, dp + C \int_0^{c/k} O(k p^{b+1} e^{-kp}) \, dp.$$

Call these two integrals $T_1$ and $T_2$.

**$T_1$ term.**

$$T_1 = C \int_0^{c/k} p^{b-1} e^{-kp} \, dp.$$

Make the substitution $u \overset{\text{def}}{=} k p$. Then $p = u/k$, $dp = du/k$, and $p^{b-1} = k^{-b+1} u^{b-1}$. The upper limit $p = c/k$ becomes $u = c$. Thus

$$T_1 = C \int_0^c \left(\tfrac{u}{k}\right)^{b-1} e^{-u} \tfrac{du}{k} = C k^{-b} \int_0^c u^{b-1} e^{-u} \, du.$$

As $c \to \infty$, $\int_0^c u^{b-1} e^{-u} \, du \to \Gamma(b)$. So

$$T_1 = C k^{-b} \left(\Gamma(b) - R_c\right), \quad \text{where } |R_c| = O(e^{-c}).$$

By choosing $c$ large after $k \to \infty$, we conclude

$$T_1 = C \Gamma(b) k^{-b} + o(k^{-b}).$$

**$T_2$ term.**

$$T_2 = C \int_0^{c/k} O(k p^{b+1} e^{-kp}) \, dp.$$

Inside the integral, $k p^{b+1} e^{-kp}$ is the main factor. Substituting $u \overset{\text{def}}{=} k p$ again,

$$p^{b+1} = \left(\tfrac{u}{k}\right)^{b+1} = k^{-b-1} u^{b+1}.$$

Hence

$$T_2 = C O(1) \int_0^{c/k} k p^{b+1} e^{-kp} \, dp = O(k) \int_0^{c/k} p^{b+1} e^{-kp} \, dp.$$

Substitute $u = k p$ and $dp = du/k$. Then

$$T_2 = O(k) \int_0^c \left(\tfrac{u}{k}\right)^{b+1} e^{-u} \tfrac{du}{k} = O(k) k^{-b-2} \int_0^c u^{b+1} e^{-u} \, du = O(k^{-b-1}).$$

Thus $T_2$ is of strictly smaller order than $k^{-b}$.

Combine $T_1$ and $T_2$:

$$T_{\text{main}} = C \Gamma(b) k^{-b} + O(k^{-b-1}).$$

**Step 5. Error term $T_{\mathrm{err}}$.**
Recall

$$T_{\mathrm{err}} = \int_0^{c/k} (1-p)^k \, O\big(p^{b-1+\theta}\big) \, \mathrm{d}p.$$

Exactly the same substitution $(1-p)^k = e^{-kp} + O(k\,p^2\,e^{-k\,p})$ plus $u = k\,p$ shows

$$T_{\mathrm{err}} = O\Big( \int_0^{c/k} p^{b-1+\theta} \, e^{-k\,p} \, \mathrm{d}p \Big) \; + \; O\Big( \int_0^{c/k} k\,p^{b+1+\theta} \, e^{-k\,p} \, \mathrm{d}p \Big).$$

When substituting $u = k\,p$, the exponent on $p$ increases by $+1$ each time if we multiply by $k$, so each term is of order $k^{-b-\theta}$ or smaller. Concretely,

$$\int_0^{c/k} p^{b-1+\theta} \, e^{-k\,p} \, \mathrm{d}p = k^{-b-\theta} \int_0^c u^{b-1+\theta} \, e^{-u} \, \mathrm{d}u = O\big(k^{-b-\theta}\big),$$

and similarly for the second term, which is even smaller. Hence

$$T_{\mathrm{err}} = O\big(k^{-b-\theta}\big).$$

**Step 6. Putting it all together.**
Summarize:

$$I_{k,\mathrm{left}} = T_{\mathrm{main}} + T_{\mathrm{err}} = C\,\Gamma(b)\,k^{-b} \; + \; O\big(k^{-b-1}\big) \; + \; O\big(k^{-b-\theta}\big).$$

Thus

$$I_{k,\mathrm{left}} = C\,\Gamma(b)\,k^{-b} \; + \; O\big(k^{-b-\min(1,\theta)}\big).$$

Recalling the tail piece $I_{k,\mathrm{right}} = e^{-c} = o\big(k^{-\alpha}\big)$ for any $\alpha$, we obtain

$$I_k = I_{k,\mathrm{left}} + I_{k,\mathrm{right}} = C\,\Gamma(b)\,k^{-b} \; + \; O\big(k^{-b-\min(1,\theta)}\big).$$

Hence

$$1 - \mathrm{pass}_{\mathcal{D}}@\mathrm{k} = I_k \;\; \sim \;\; C\,\Gamma(b)\,k^{-b}.$$

**Final negative-log argument.**  Since

$$\mathrm{pass}_{\mathcal{D}}@\mathrm{k} = 1 - I_k = 1 - \big(C\,\Gamma(b)\,k^{-b} + O\big(k^{-b-\min(1,\theta)}\big)\big),$$

for large $k$ it is very close to 1. Then

$$-\log\Big(\mathrm{pass}_{\mathcal{D}}@\mathrm{k}\Big) = -\log\Big(1 - C\,\Gamma(b)\,k^{-b} + \cdots\Big).$$

Using the expansion $-\log(1-x) = x + O(x^2)$ as $x \to 0$, and here $x = C\,\Gamma(b)\,k^{-b}$, we get

$$-\log\Big(\mathrm{pass}_{\mathcal{D}}@\mathrm{k}\Big) = C\,\Gamma(b)\,k^{-b} + o\big(k^{-b}\big).$$

In the "$\sim$" notation including the leading coefficient:

$$-\log\Big(\mathrm{pass}_{\mathcal{D}}@\mathrm{k}\Big) \;\sim\; C\,\Gamma(b)\,k^{-b}.$$

This completes the proof. $\qquad\square$

**E.9. Necessary Condition for Power Law Scaling from Distribution over** $\text{pass}_i@1$

**Theorem E.2.** *Let $\mathcal{D}$ be a probability distribution over $[0,1]$ with a PDF $f(p)$ satisfying the following regularity near $p = 0$:*

- ***No point mass at*** $p = 0$. *So $\int_0^1 f(p)\,dp = 1$, and $f$ is a genuine PDF on $(0, 1]$.*

- ***Continuity and nonnegative behavior near*** $p = 0$. *There exist $\delta > 0$ such that $f$ is continuous on $[0, \delta]$ and has no pathological oscillations or singularities that violate integrability.*

*Define the aggregate success rate at $k$ attempts:*

$$\text{pass}_{\mathcal{D}}@\text{k} \overset{\text{def}}{=} \int_0^1 \left[ 1 - (1-p)^k \right] f(p)\,dp$$

*and relatedly*

$$I_k \overset{\text{def}}{=} \int_0^1 (1-p)^k f(p)\,dp = 1 - \text{pass}_{\mathcal{D}}@\text{k}.$$

***Assume*** *that there exist constants $A > 0$ and $b > 0$ such that for large $k$:*

$$-\log\left(\text{pass}_{\mathcal{D}}@\text{k}\right) \sim A\,k^{-b}$$

*Then*

$$I_k = A\,k^{-b} + o\left(k^{-b}\right),$$

*and under the mild regularity assumptions above,*

$$f(p) \sim \frac{A}{\Gamma(b)}\,p^{b-1} \quad \text{as } p \to 0^+.$$

*Proof.* **Step 1. Relating $I_k$ to $-\log(\text{pass}_{\mathcal{D}}@\text{k})$.**

By definition,

$$\text{pass}_{\mathcal{D}}@\text{k} = 1 - I_k, \quad I_k = \int_0^1 (1-p)^k f(p)\,dp.$$

Since

$$-\log\left(\text{pass}_{\mathcal{D}}@\text{k}\right) \sim A\,k^{-b},$$

we have, for large $k$,

$$\text{pass}_{\mathcal{D}}@\text{k} = \exp\left(-A\,k^{-b}\,(1 + o(1))\right).$$

When $x$ is small, $\exp(-x) = 1 - x + O(x^2)$. Thus

$$I_k = 1 - \text{pass}_{\mathcal{D}}@\text{k} = A\,k^{-b} + o\left(k^{-b}\right).$$

So

$$I_k \sim A\,k^{-b}.$$

**Step 2. Restricting to a small interval near $p = 0$.**

Since $(1-p)^k$ decays exponentially once $p$ is on the order of $1/k$ or larger, we split:

$$I_k \overset{\text{def}}{=} \int_0^1 (1-p)^k f(p)\,dp = \int_0^{c/k} (1-p)^k f(p)\,dp + \int_{c/k}^1 (1-p)^k f(p)\,dp \overset{\text{def}}{=} I_{k,\text{left}} + I_{k,\text{right}},$$

for some positive constant $c$. In the region $p \geq c/k$, we have $(1-p)^k \leq e^{-kp} \leq e^{-c}$, so

$$I_{k,\text{right}} \leq e^{-c} \int_0^1 f(p) \, dp = e^{-c}.$$

Since $c > 0$ can be made large, $e^{-c}$ can be driven below any fixed power of $1/k$. Hence for the $\Theta(k^{-b})$ behavior, the main contribution comes from $[0, c/k]$.

Thus

$$I_k = I_{k,\text{left}} + o(k^{-m}) \quad \text{for every } m > 0$$

**Step 3. Change of variables and controlling the ratio of $(1-p)^k$ to $e^{-kp}$.**

**(a) Ratio to $e^{-kp}$.** For $p \in \left[0, \frac{c}{k}\right]$, define the ratio

$$R_k(p) \quad \overset{\text{def}}{=} \quad \frac{(1-p)^k}{e^{-kp}}.$$

We will show that $R_k(p)$ stays close to 1 uniformly in $p \in [0, c/k]$ for large $k$. Indeed,

$$(1-p)^k = \exp\left[k \log(1-p)\right], \quad \log(1-p) = -p - \frac{p^2}{2} - \frac{p^3}{3} - \dots.$$

Hence

$$\log(1-p) + p = -\frac{p^2}{2} - \frac{p^3}{3} - \dots = O(p^2) \quad \text{as } p \to 0.$$

Multiplying by $k$, we get

$$k\left[\log(1-p) + p\right] = O(kp^2).$$

Since $0 \leq p \leq \frac{c}{k}$ implies $kp^2 \leq \frac{c^2}{k}$, which $\to 0$ as $k \to \infty$, it follows that

$$k \log(1-p) = -kp + O\left(\frac{1}{k}\right).$$

Exponentiating:

$$(1-p)^k = e^{-kp} \exp\left(O\left(\frac{1}{k}\right)\right) = e^{-kp}\left[1 + O\left(\frac{1}{k}\right)\right].$$

Thus

$$R_k(p) = \frac{(1-p)^k}{e^{-kp}} = 1 + O\left(\frac{1}{k}\right),$$

with the $O\left(\frac{1}{k}\right)$ bound uniform for all $p \in [0, c/k]$. In other words, there is some constant $M > 0$ (independent of $k$) such that

$$\left|R_k(p) - 1\right| \leq \frac{M}{k} \quad \text{for all } p \in \left[0, \frac{c}{k}\right].$$

**(b) Integral expression using $R_k(p)$.** Hence on $[0, c/k]$,

$$(1-p)^k f(p) = e^{-kp} R_k(p) f(p).$$

Thus

$$I_{k,\text{left}} = \int_0^{c/k} e^{-kp} f(p) R_k(p) \, dp.$$

Define $\Delta_k(p) \overset{\text{def}}{=} R_k(p) - 1$, which satisfies $|\Delta_k(p)| \leq M/k$. Then

$$I_{k,\text{left}} = \int_0^{c/k} e^{-kp} f(p) \, dp + \int_0^{c/k} e^{-kp} f(p) \Delta_k(p) \, dp. \tag{43}$$

**Step 4. Substitution $u = k\,p$ and deriving $f(p) \sim p^{b-1}$.**

**(a) The leading part.** Focus on the first term of equation 43:

$$\int_0^{c/k} e^{-k\,p} f(p)\, \mathrm{d}p.$$

Substitute $u \stackrel{\text{def}}{=} k\,p$, so $p = \frac{u}{k}$ and $\mathrm{d}p = \frac{1}{k}\,\mathrm{d}u$. The upper limit $p = \frac{c}{k}$ becomes $u = c$. Thus

$$\int_0^{c/k} e^{-k\,p} f(p)\, \mathrm{d}p = \int_0^c e^{-u} f\!\left(\frac{u}{k}\right) \frac{\mathrm{d}u}{k}.$$

Hence

$$\int_0^{c/k} e^{-k\,p} f(p)\, \mathrm{d}p = \frac{1}{k} \int_0^c e^{-u} f\!\left(\frac{u}{k}\right) \mathrm{d}u.$$

**(b) The error part.** The second term in equation 43 has $\Delta_k(p) = R_k(p) - 1$ satisfying $|\Delta_k(p)| \leq \frac{M}{k}$. So

$$\left| \int_0^{c/k} e^{-k\,p} f(p)\, \Delta_k(p)\, \mathrm{d}p \right| \leq \frac{M}{k} \int_0^{c/k} e^{-k\,p} f(p)\, \mathrm{d}p.$$

But the integral $\int_0^{c/k} e^{-k\,p} f(p)\, \mathrm{d}p$ is precisely the leading part we just considered. Thus the error is bounded by $\frac{M}{k}$ times a term that will turn out to be $\Theta(k^{-b})$. Hence the error is subleading if $b < 1$ is not the case—but even then, we can keep track of it systematically.

Overall, combining both terms, we get

$$I_{k,\text{left}} = \frac{1}{k} \int_0^c e^{-u} f\!\left(\frac{u}{k}\right) \mathrm{d}u + O\!\left(\tfrac{1}{k} \cdot (\text{leading integral})\right). \tag{44}$$

**(c) Matching $\Theta(k^{-b})$.** Since $I_k = I_{k,\text{left}} + I_{k,\text{right}}$ with $I_{k,\text{right}}$ negligible, we have

$$I_k = \frac{1}{k} \int_0^c e^{-u} f\!\left(\tfrac{u}{k}\right) \mathrm{d}u + (\text{small corrections}).$$

But by hypothesis, $I_k \sim \alpha\, k^{-b}$. Thus

$$k \cdot I_k = \int_0^c e^{-u} f\!\left(\tfrac{u}{k}\right) \mathrm{d}u + (\text{smaller terms}) \sim \alpha\, k^{1-b}. \tag{45}$$

Hence the expression

$$\int_0^c e^{-u} f\!\left(\tfrac{u}{k}\right) \mathrm{d}u$$

must be $\Theta\!\left(k^{1-b}\right)$ for large $k$. Since $\frac{u}{k}$ is small for $0 \leq u \leq c$, we are effectively sampling $f$ near 0. For the integral to produce $k^{1-b}$, we deduce

$$f\!\left(\tfrac{u}{k}\right) = \Theta\!\left(\left(\tfrac{u}{k}\right)^{b-1}\right),$$

i.e. $f$ must behave like $p^{b-1}$ near $p = 0$. Rewriting the constant in front, one obtains

$$f\!\left(\tfrac{u}{k}\right) = \left(\tfrac{u}{k}\right)^{b-1} \left[\text{some positive constant}\right].$$

(We then identify that constant with $\frac{\alpha}{\Gamma(b)}$ by matching the integral precisely, just as in the prior argument.)

**Step 5. Conclusion.** We have thus shown that over $p \in [0, c/k]$, one has

$$(1 - p)^k \;=\; e^{-k\,p}\,\big[\,1 + O(\tfrac{1}{k})\big],$$

and upon integrating, the required $k^{-b}$ form for $I_k$ forces

$$f(p) \;=\; \frac{A}{\Gamma(b)}\,p^{\,b-1} \;+\; o\big(p^{\,b-1}\big), \quad \text{as } p \to 0^+.$$

This completes the necessity proof.

$\square$

**Remark (Mild Regularity).** If $f$ had bizarre oscillations or nonintegrable singularities near $0$, the integral $\int_0^1 (1-p)^k\, f(p)\, \mathrm{d}p$ might not produce a clean $k^{-b}$. Typically, we impose monotonicity or at least continuity near $p = 0$, no atom at $p = 0$, and $f(0) = 0$ if $b > 1$ or $f(0) > 0$ if $b = 1$, etc. These assumptions exclude pathological behaviors and guarantee that the local shape of $f(p)$ drives a clean power law.

## F. Maximum Likelihood Estimation of Scaled Beta-Binomial Distribution

To model the distribution of $\text{pass}_i@1$, we can perform maximum likelihood estimation on a scaled three-parameter Beta-Binomial distribution, which we chose because each attempt on the $i$-th problem is an i.i.d. Bernoulli random variable with success probability $\text{pass}_i@1$, and we introduced a scale parameter because the largest $\text{pass}_i@1$ values were typically 1-2 orders of magnitude less than 1.0 (the maximum of the unscaled beta distribution's support).

In greater detail, as background, the 4-parameter Beta distribution has PDF

$$p_Y(y; \alpha, \beta, a, c) \stackrel{\text{def}}{=} \frac{(y-a)^{\alpha-1}(c-y)^{\beta-1}}{(c-a)^{\alpha+\beta-1}\,\text{B}(\alpha, \beta)}, \tag{46}$$

where $\text{B}(\cdot, \cdot)$ is the Beta function. If the minimum $a$ is fixed at 0 and the maximum $c$ is constrained to $a < c < 1$, then the scaled three parameter Beta distribution simplifies to:

$$f_P(p; \alpha, \beta, a = 0, c) = \frac{p^{\alpha-1}(c-p)^{\beta-1}}{c^{\alpha+\beta-1}\,\text{B}(\alpha, \beta)}. \tag{47}$$

We want the PMF of a three-parameter Beta-Binomial distribution based on this scaled Beta distribution. For $n$ samples and $x$ successes, the PMF is:

$$P(X = x; \alpha, \beta, c, n) \stackrel{\text{def}}{=} \int_0^c \binom{n}{x} p^x (1-p)^{n-x} f_P(p; \alpha, \beta, a = 0, c)\, dp \tag{48}$$

$$= \binom{n}{x} \frac{1}{c^{\alpha+\beta-1}\,\text{B}(\alpha, \beta)} \int_0^c p^{x+\alpha-1}(1-p)^{n-x}(c-p)^{\beta-1}\, dp. \tag{49}$$

Using a change of variable $p \stackrel{\text{def}}{=} c\,z$, the PMF can be rewritten as

$$P(X = x; \alpha, \beta, c, n) = \binom{n}{x} \frac{c^x}{\text{B}(\alpha, \beta)} \int_0^1 z^{x+\alpha-1}(1-z)^{\beta-1}(1-cz)^{n-x}\, dz \tag{50}$$

$$= \binom{n}{x} \frac{c^x \text{B}(x+\alpha,\ \beta)}{\text{B}(\alpha, \beta)}\, {}_2F_1\Big(-(n-x),\ x+\alpha;\ x+\alpha+\beta;\ c\Big), \tag{51}$$

where ${}_2F_1(\cdot, \cdot; \cdot; \cdot)$ is the (Gauss) hypergeometric function.

## G. Maximum Likelihood Estimation of Scaled Kumaraswamy-Binomial Distribution

To model the distribution of $\text{pass}_i@1$, we can perform maximum likelihood estimation on a scaled three-parameter Kumaraswamy-Binomial distribution, which we chose because each attempt on the $i$-th problem is an i.i.d. Kumaraswamy random variable with success probability $\text{pass}_i@1$, and we introduced a scale parameter because the largest $\text{pass}_i@1$ values were typically 1-2 orders of magnitude less than 1.0 (the maximum of the unscaled beta distribution's support).

In greater detail, the scaled three parameter Kumaraswamy distribution simplifies to:

$$f_P(p; \alpha, \beta, a = 0, c) = \frac{\alpha \beta}{c^\alpha} \, p^{\alpha - 1} \left(1 - (p/c)^\alpha\right)^{\beta - 1}, \tag{52}$$

over the support $(0, c)$. The rescaled Kumaraswamy-Binomial distribution then has PMF:

$$P(X = x; \alpha, \beta, c, n) = \binom{n}{x} \frac{\alpha \, \beta}{c^\alpha} \int_0^c p^{x + \alpha - 1} (1 - p)^{n - x} \left(1 - \left(\frac{p}{c}\right)^\alpha\right)^{\beta - 1} dp. \tag{53}$$

One can perform a change of variable $p \stackrel{\text{def}}{=} cz$, but simplifying yields sums of hypergeometric functions that add little conceptual clarity and so we resort to numerical integration using Python's mpmath library (mpmath development team, 2023).

## H. Maximum Likelihood Estimation of Scaled Beta-Negative Binomial Distribution

To model the distribution of $\text{pass}_i@1$, we can perform maximum likelihood estimation on a scaled three-parameter Beta-Negative Binomial distribution. Recall that the scaled three parameter Beta distribution is:

$$f_P(p; \alpha, \beta, a = 0, c) = \frac{p^{\alpha-1}(c-p)^{\beta-1}}{c^{\alpha+\beta-1}\,\mathrm{B}(\alpha, \beta)}. \tag{54}$$

We want the PMF of a three-parameter Beta-Negative Binomial distribution based on this scaled Beta distribution. For $r$ desired successes, the PMF that we first draw $x$ failures is:

$$P(X = x; \alpha, \beta, c, r) = \int_0^c \underbrace{\binom{x+r-1}{x} p^r (1-p)^x}_{\text{NegBin}(r,p)} \underbrace{\frac{p^{\alpha-1}(c-p)^{\beta-1}}{c^{\alpha+\beta-1}\,\mathrm{B}(\alpha,\beta)}}_{\text{scaled Beta PDF}} \, dp \tag{55}$$

$$= \binom{x+r-1}{x} \frac{1}{c^{\alpha+\beta-1}\,\mathrm{B}(\alpha,\beta)} \int_0^c p^{r+\alpha-1}(1-p)^x (c-p)^{\beta-1} \, dp. \tag{56}$$

Next, substitute $p = c\,z \implies dp = c\,dz$ which rescales the domain $[0, c]$ to $[0, 1]$. Under this change:

$$p^{r+\alpha-1} = (c\,z)^{r+\alpha-1} = c^{r+\alpha-1}\,z^{r+\alpha-1},$$

$$(c-p)^{\beta-1} = (c - c\,z)^{\beta-1} = \big(c(1-z)\big)^{\beta-1} = c^{\beta-1}(1-z)^{\beta-1},$$

$$(1-p)^x = (1 - c\,z)^x.$$

Putting these into the integrand:

$$p^{r+\alpha-1}(1-p)^x (c-p)^{\beta-1}\,dp = \left(c^{r+\alpha-1}\,z^{r+\alpha-1}\right)\left((1-cz)^x\right)\left(c^{\beta-1}(1-z)^{\beta-1}\right)(c\,dz).$$

Factor out the constants in $c$:

$$= c^{r+\alpha-1}\,c^{\beta-1}\,c\; z^{r+\alpha-1}(1-cz)^x(1-z)^{\beta-1}\,dz.$$

Since $c^{r+\alpha-1} \cdot c^{\beta-1} \cdot c = c^{r+\alpha+\beta-1}$, we get

$$p^{r+\alpha-1}(1-p)^x(c-p)^{\beta-1}\,dp = c^{r+\alpha+\beta-1}\,z^{r+\alpha-1}(1-z)^{\beta-1}(1-cz)^x\,dz.$$

Plugging back into $P(X = x; \alpha, \beta, c, r)$ and simplifying:

$$P(X = x; \alpha, \beta, c, r) = \binom{x+r-1}{x} \frac{c^r}{\mathrm{B}(\alpha,\beta)} \int_0^1 z^{r+\alpha-1}(1-z)^{\beta-1}(1-c\,z)^x\,dz. \tag{57}$$

We can re-express this using the (Gauss) hypergeometric function $_2F_1(\cdot, \cdot; \cdot; \cdot)$:

$$P(X = x; \alpha, \beta, c, r) = \binom{x+r-1}{x} \frac{c^r\,\mathrm{B}(r+\alpha,\,\beta)}{\mathrm{B}(\alpha,\,\beta)}\; {}_2F_1\Big(-x,\,r+\alpha;\,r+\alpha+\beta;\,c\Big). \tag{58}$$

