# OpenReview forum: "How Do Large Language Monkeys Get Their Power (Laws)?"
_ICML.cc/2025/Conference — ICML 2025 oral_

### Official Review · Reviewer_wsdP · 2025-03-06

**Overall Recommendation:** 4

**Summary:**

This work tries to explain a curious phenomenon in LLM test-time scaling via repeated sampling and verification, as well as in Best-of-N jailbreaking:
while the per-problem failure probability should decay exponentially with the number of attempts,
it is often observed in practice that the average success rate on a task (which contains multiple problems) exhibits a power law instead.
The authors prove theoretically that this should be the case if (and only if) the distribution of per-problem single-attempt success probability satisfies certain long-tailed property;
such a condition is validated empirically for multiple tasks and LLMs.
Based on such analyses, this work also proposes a distributional estimator for the coefficients in the power laws of repeated sampling, and validates its efficacy numerically.

**Update after rebuttal:** I have read the authors' rebuttal (as well as other reviews), and will maintain my positive evaluation.

**Claims And Evidence:**

The claims made in the submission are supported by clear and convincing evidence, both theoretically and numerically.

**Essential References Not Discussed:**

Not that I'm aware of.

**Experimental Designs Or Analyses:**

I have checked all experimental designs and analyses, which are mostly standard statistical analyses for supporting the developed theories.
I don't see any serious issue in the results.

**Methods And Evaluation Criteria:**

The proposed methods and evaluation criteria make sense to me.

**Other Comments Or Suggestions:**

A typo in Line 413 Right: "contributes is a new hypothesis" --> "contributes a new hypothesis"

**Other Strengths And Weaknesses:**

N/A

**Questions For Authors:**

N/A

**Relation To Broader Scientific Literature:**

This work offers some mathematical insights for LLM inference scaling laws that have been extensively studied recently.
The key to solving the puzzle under consideration can be easily explained in one sentence (in Line 231 Left):
"(a known result that) power laws can originate from an appropriately weighted sum of exponential functions".
Although the solution becomes obvious once it has been presented,
it is the essential contribution of this work to bring this to light.

**Theoretical Claims:**

I read through the theoretical analyses in the main text, which make sense to me.
I only skimmed through the technical proofs in the appendix.

---

> ### Author Rebuttal · Authors · 2025-03-31
>
> Thank you for your positive and thorough review of our work. We appreciate your thoughtful assessment of our theoretical and numerical analyses, as well as your recognition of our contribution in applying the mathematical insight about power laws emerging from weighted sums of exponential functions to this specific domain. We will certainly fix the identified typo in Line 413, changing "contributes is a new hypothesis" to "contributes a new hypothesis."
>
> We are committed to making this paper as strong as possible and would value your guidance on what specific improvements would strengthen the manuscript further. If you have any additional suggestions that would elevate your assessment to a 'Strong Accept,' we would be grateful for that feedback and would make every effort to address those points in our revision.
>
> Thank you again for your constructive engagement with our work.

---

### Official Review · Reviewer_oCti · 2025-03-15

**Overall Recommendation:** 4

**Summary:**

The paper demonstrates that power law behaviour in “pass at k” metrics originates from a power law tail in the distribution of the “pass at 1” probability across the test set. Furthermore, it argues that directly modeling the “pass at 1” distribution leads to more accurate predictions for the values of “pass at k.”

## Update after rebuttal

I maintain my positive assessment of the paper.

**Claims And Evidence:**

Yes.

**Essential References Not Discussed:**

I am not aware of any gross omission of related work, but I am also not satisfied with the related work section in the submitted paper - see “Comments Or Suggestions” below.

**Experimental Designs Or Analyses:**

I have checked the experimental design and analysis to the level with which it is described in the main text. Overall, I found it satisfactory; see a question and a suggestion below.

**Methods And Evaluation Criteria:**

Yes.

**Other Comments Or Suggestions:**

1. While overall the paper is well written, Section 6 (Related Work) is sub-par: it is a single paragraph spanning over more than a column that reads as a laundry list of papers about scaling laws. I can find better lists of this sort online. What I expect to find in a related work section is insight about how these works relate to the paper. Here specifically, I am missing a discussion about prior work attempting to explain the origin of scaling laws currently listed in lines 390 to 395 - do any of these models predict a power law tail for the distribution of difficulties of individual test data? Could they provide a parametric form for that distribution? Polo et al. (2024) likely also deserve more detailed discussion due to the pass@k experiments they describe in Section 4.5.

2. Figure 4 is missing some indication of what is the measurement error of pass@1. I am not sure what is the best way to visualize it - at the very least, you should indicate the (inverse of the) sample size used to obtain the pass@1 estimates.

**Other Strengths And Weaknesses:**

Covered by my answers to the other questions.

**Questions For Authors:**

Is the discretize+ML described in lines 370-379 optimal in any sense? More specifically, is it the true maximum likelihood estimator of the pass@1 distribution parameters given the observations?

**Relation To Broader Scientific Literature:**

This paper provides a simple but valuable insight into the shape of scaling laws with respect to success probability after multiple attempts. Prior work observed these sometimes behave as a power law, but this appears to be the first work to reconcile this finding with the fact that for an individual data point, the pass at k probability must decay exponentially.

**Theoretical Claims:**

I did not check the proofs and derivations in the appendix, but overall the analytical claims in the paper make sense and agree with similar calculations I have made in the past in different contexts.

---

> ### Author Rebuttal · Authors · 2025-03-31
>
> We appreciate your constructive feedback. We address your points below.
>
> ### Improvements to Related Work
>
> >  Could they provide a parametric form for that distribution? Polo et al. (2024) likely also deserve more detailed discussion due to the pass@k experiments they describe in Section 4.5.
>
> We will incorporate a more thorough discussion of Polo et al. (2024)'s Sloth and the related papers Ruan et al. (2024)'s Observational Scaling Laws, and Owen (2024)'s predictability analysis. Their latent variable regression models offer complementary perspectives to our approach. Interestingly, Polo's Section 4.5 pass@k experiments show relatively poor fits (their Figure 7), possibly because (as best as we can tell) they use the biased estimator of $\operatorname{pass_i@k}$ that Chen et al. (2021) caution against. That said, while Polo et al. (2024) don’t define a functional form for scaling, our mathematical analysis could be combined with their estimated per-problem single-attempt success probabilities $\operatorname{pass_i@1}$. It’s possible that their cross-benchmark fitting method gives better estimates of these per-problem single-attempt success probabilities, which would improve our method.
>
> We will expand on this connection and highlight how per-problem analyses could potentially combine with such cross-benchmark approaches to further improve predictability.
>
> > What I expect to find in a related work section is insight about how these works relate to the paper. Here specifically, I am missing a discussion about prior work attempting to explain the origin of scaling laws currently listed in lines 390 to 395 - do any of these models predict a power law tail for the distribution of difficulties of individual test data?
>
> We will revise the related work section to discuss contributions from key works beyond just listing scaling law analyses. To the best of our knowledge, no prior work has specifically attempted to explain the power law emergence with repeat sampling in the manner we propose, perhaps due to the recency of works like Brown et al. (2024) and Hughes et al. (2024).
>
> ### Quantification of Measurement Error
>
> This is an excellent suggestion. For Large Language Monkeys, each problem had 10,000 attempts sampled, making the per-problem single-attempt success rate $\operatorname{pass_i@1}$ a Bernoulli estimator with well-understood standard error. The Best-of-N jailbreak case is more nuanced due to varying sample sizes across problems (as detailed in Appendix A).
>
> We will add this measurement precision information to the main text and try to develop an appropriate visualization to represent the uncertainty in Figure 4.
>
> ### Optimality of Discretized ML Estimator
>
> > Is the discretize+ML described in lines 370-379 optimal in any sense? More specifically, is it the true maximum likelihood estimator of the pass@1 distribution parameters given the observations?
>
> Regarding whether our discretize+ML approach is optimal: we cannot make such a strong claim. While our empirical results demonstrate its effectiveness for the specific task of power law exponent estimation, a formal proof of optimality would require additional theoretical analysis.
>
> The approach may be particularly well-suited for estimating parameters that best describe the distribution's left tail, which is crucial for our application. However, as you correctly suggest, this is a potentially complex statistical estimation question that warrants dedicated investigation. We will clarify these limitations in our revision and position this as an opportunity for future research.
>
> ### Invitation for Ways to Improve
>
> We believe addressing these points will substantially improve the paper's clarity and impact.
>
> We are committed to making this paper as strong as possible and would value your guidance on what specific improvements would strengthen the manuscript further. If you have any additional suggestions that would elevate your assessment to a 'Strong Accept,' we would be grateful for that feedback and would make every effort to address those points in our revision.

---

> > ### Comment · Reviewer_oCti · 2025-04-06
> >
> > Thank you for the detailed response. I maintain my current positive assessment.

---

### Official Review · Reviewer_5fN8 · 2025-03-17

**Overall Recommendation:** 4

**Summary:**

This paper explores the scaling behavior of LLMs when inference-time compute is increased through repeated sampling. While failure rates for individual problems should decrease exponentially with multiple attempts, the authors observe that the aggregate success rate across many problems follows a power law. They resolve this paradox by demonstrating that this phenomenon arises from the distribution of single-attempt success probabilities, which is heavy-tailed that a small fraction of extremely difficult tasks disproportionately influences the overall trend. Through empirical analysis on math problem-solving and multimodal jailbreaking tasks, they confirm that while individual tasks improve exponentially, the global trend follows a power law due to the nature of the problem distribution. This work introduces a new distributional estimator that predicts the power law exponent with far less compute than traditional methods, improving efficiency by 2-4 orders of magnitude. Furthermore, they explain why some models deviate from power law scaling, attributing it to the lack of a heavy-tailed success rate distribution. Ultimately, this research enhances the understanding of inference-time scaling and provides a more accurate framework for forecasting LLM performance, offering practical implications for model evaluation and optimization.

## update after rebuttal

I appreciate the clarification and insight from the authors. My concerns have been addressed. I have updated the scores accordingly.

**Claims And Evidence:**

### Theoretical Justification for Heavy-Tailed Distributions:
While the authors demonstrate empirically that single-attempt success rates follow a heavy-tailed distribution, they do not provide a deeper theoretical justification for why this occurs in practice. They speculate that benchmark design and selection bias may contribute, but these points are not rigorously analyzed.

**Essential References Not Discussed:**

N/A

**Experimental Designs Or Analyses:**

The experimental design is well-structured to investigate the scaling behavior of LLMs under inference-time compute scaling. The authors conduct two core experiments: mathematical problem-solving using Pythia models on the MATH benchmark and Best-of-N jailbreaking on HarmBench, analyzing how success rates improve with multiple attempts. The negative log success rate (-log(pass@k)) is used as a primary metric, effectively distinguishing between exponential and power law scaling trends. Additionally, the authors fit success rate distributions using Beta and Kumaraswamy distributions, demonstrating that the heavy-tailed nature of single-attempt success probabilities explains power law behavior. Their proposed distributional estimator for power law exponents significantly reduces compute requirements by 2-4 orders of magnitude, showing clear advantages over traditional methods.

However, the study is somewhat limited in scope, as the sample sizes for both benchmarks (128 math problems, 159 jailbreaking prompts) may not fully capture model behavior across diverse tasks.

**Methods And Evaluation Criteria:**

Yes. The paper employs a rigorous mathematical framework to establish that while individual problems exhibit exponential failure rate decay, aggregate success rates follow a power law due to the heavy-tailed distribution of single-attempt success probabilities. Empirical validation is conducted on two key tasks: mathematical problem-solving using the MATH benchmark and multimodal jailbreaking using HarmBench, both of which effectively illustrate how repeated sampling impacts model performance.

The paper also introduces a distributional estimator that predicts power law scaling exponents more efficiently than traditional regression-based methods, reducing computational costs by 2-4 orders of magnitude. The evaluation criteria, particularly the use of negative log success rate (−log(pass@k)), are well-motivated and provide clear insights into model scaling behavior. However, while the chosen benchmarks are appropriate, the study does not explore whether similar scaling laws hold across a broader range of NLP tasks such as summarization or question answering. Additionally, the underlying causes of heavy-tailed success probability distributions remain speculative.

**Other Comments Or Suggestions:**

N/A

**Other Strengths And Weaknesses:**

### Strengths

- The paper presents a novel theoretical framework explaining why per-problem success rates follow exponential decay, while aggregate success rates exhibit power law behavior due to heavy-tailed task difficulty distributions.

- The introduction of a distributional estimator for power law exponents is innovative and significantly improves compute efficiency, making it a practical contribution for LLM evaluation.

- The work has security implications, particularly in adversarial robustness and jailbreaking prevention, by explaining how increased attack attempts affect model vulnerabilities.

- The paper is well-organized, with a clear presentation of mathematical derivations and strong empirical validation.

### Weaknesses

- The experiments focus on MATH (128 problems) and HarmBench (159 adversarial prompts), which may not fully generalize to other NLP tasks (e.g., summarization, question answering, commonsense reasoning).

- The proposed estimator for power law exponents is validated on synthetic data and limited benchmarks, but its performance on real-world applications (e.g., machine translation, conversational AI) remains uncertain.

**Questions For Authors:**

The study notes that some models (e.g., LLaMA 3 8B IT) do not follow power law scaling. Do you have hypotheses on why these models deviate from the expected trend? Could this be due to model architecture, training objectives, or tokenization differences? Understanding these deviations would help clarify when power law inference-time scaling can and cannot be expected.

**Relation To Broader Scientific Literature:**

This work builds upon and extends key areas in the scientific literature on scaling laws, inference-time compute strategies, and power law behaviors in LLMs. It connects to scaling laws which showed that model performance improves predictably with increased compute, data, and parameter count, and later being refined by emphasized data efficiency over sheer model size. However, while previous work primarily focused on pretraining compute scaling, this paper shifts the focus to inference-time compute scaling, showing how repeated sampling affects model success rates. The discovery that per-problem failure rates decreases exponentially while aggregate success follows a power law introduces a new perspective, linking task difficulty distributions to inference efficiency.

Additionally, the study relates to Best-of-N sampling strategies which demonstrated that generating multiple outputs and selecting the best significantly improves performance. This paper extends those insights by providing a theoretical framework explaining why repeated inference exhibits power law scaling, depending on the distribution of single-attempt success probabilities.

**Theoretical Claims:**

The key theoretical contributions involve proving that per-problem failure rates decay exponentially while the aggregate success rate across problems follows a power law due to the distributional properties of single-attempt success rates.

- 1. Exponential Decay of Per-Problem Failure Rates
   - Claim: If each attempt at solving a problem is independent with a fixed success probability, then the failure probability over $k$ attempts follows an exponential decay.
   - This confirms that the failure rate decreases exponentially as $k$ increases. The proof is valid and follows standard probability theory, particularly the Bernoulli trial model where repeated independent attempts lead to geometric or exponential-like decay.

- 2. Power Law Scaling of Aggregate Success Rates
   - Claim: Despite individual problems following exponential failure rate decay, the overall success rate across problems follows a power law if the distribution of single-attempt success probabilities is heavy-tailed.
   - The authors show that if the distribution $p_D(pass_i@1)$ has a power-law-like left tail near zero, the resulting negative log success rate follows a power law in $k$. They provide sufficiency and necessity theorems, proving that this scaling occurs if and only if the distribution of $pass_i@1$ behaves in a certain way. The derivation follows known statistical results about sums of exponentials forming power laws in appropriate conditions. The use of Gamma functions and integral approximations aligns with established results in scaling law analysis.

- 3. Connection Between Distributional Shape and Power Law Exponents
   - Claim: The power law exponent $b$ of the aggregate scaling behavior is directly determined by the shape of the distribution of single-attempt success probabilities.
    - The authors analyze different statistical distributions (e.g., Beta, Kumaraswamy, Continuous Bernoulli) and derive their impact on the resulting power law exponent. They prove that a heavy-tailed distribution of single-attempt success rates naturally leads to power law scaling. The derivations match well-known properties of compound binomial distributions, where a sum of many exponentially decaying functions with varying rates can form a power law.

---

> ### Author Rebuttal · Authors · 2025-03-31
>
> Thank you for your thoughtful review. We appreciate your recognition of our work's strengths, particularly that our paper "presents a novel theoretical framework explaining why per-problem success rates follow exponential decay, while aggregate success rates exhibit power law behavior" and that our distributional estimator "significantly improves compute efficiency, making it a practical contribution for LLM evaluation."
>
> ### Models and Tasks Are Diverse
>
> > However, the study is somewhat limited in scope, as the sample sizes for both benchmarks (128 math problems, 159 jailbreaking prompts) may not fully capture model behavior across diverse tasks.
>
> We believe our study offers substantial diversity.
>
> 1. Our analysis spans leading frontier models from four major AI companies (OpenAI, Google, Anthropic, Meta), open-parameter models ranging from 17M to 12B parameters, and fundamentally different tasks (mathematical problem solving and multimodal jailbreaking). This diversity strengthens our confidence in the generalizability of our findings.
>
> 2. We agree that verification across an even wider range of models and tasks would further strengthen generalizability. We relied on existing datasets from Brown et al. (2024) and Hughes et al. (2024) because generating 10,000+ attempts per model per problem involves substantial computational costs. For perspective, an experiment with 10 models across 5 benchmarks with 100 problems each would require 50 million sampled outputs.
>
> 3. From a statistical perspective, sample sizes matter in order to make precise statistical statements, e.g., determining confidence intervals. If you feel like 128 and 159 problems with >=10k samples per problem are inadequate for specific claims, could you please tell us which claims you find inadequately justified so we can better assess?
>
> ### Real-World Applications
>
> > The proposed estimator for power law exponents is validated on synthetic data and limited benchmarks, but its performance on real-world applications (e.g., machine translation, conversational AI) remains uncertain.
>
>
> This is a valid concern. While our current validation on both synthetic and real-world data demonstrates the estimator's effectiveness, we acknowledge the need for broader validation across diverse applications. Our method's foundations in statistical theory provide confidence in its generalizability, but we agree that testing across additional domains would be valuable. We view this as an important direction for future work and are exploring partnerships to apply our estimator to machine translation, conversational AI, and other practical applications.
>
> ### Deviations from Power Law Scaling
>
> > The study notes that some models (e.g., LLaMA 3 8B IT) do not follow power law scaling. Do you have hypotheses on why
>
> Our theoretical framework provides a clear explanation: power law scaling emerges only when the distribution of single-attempt success probabilities has a heavy left tail, which Llama 3 8B IT lacks when tested on jailbreaking (as shown in Figure 4).
>
> What this means practically is that Llama 3 8B IT has lower robustness against adversarial attacks than the other models Hughes et al. 2024 tested. This could be attributable to many reasons. This could stem from several factors, including its smaller size, potentially less extensive safety training, or absence of defense mechanisms likely present in API-based models like GPT, Claude, and Gemini. Unfortunately, the proprietary nature of these other models limits our ability to investigate these hypotheses further.
>
> ### Deeper Theoretical Analysis of Why Heavy Left Tails Appear
>
> > While the authors demonstrate empirically that single-attempt success rates follow a heavy-tailed distribution, they do not provide a deeper theoretical justification for why this occurs in practice. They speculate that benchmark design and selection bias may contribute, but these points are not rigorously analyzed.
>
> The best answer we can think of is that power law scaling emerges in a "Goldilocks zone" of problem difficulty. For heavy left-tails to appear, we need problems that are challenging but not impossible—difficult enough to require many attempts yet still solvable. This explains why we wouldn't observe power law scaling when applying state-of-the-art models like GPT-4.5 or Claude 3.7 Sonnet to relatively simple benchmarks like GLUE (too easy), nor when applying these same models to extremely difficult tasks like Millennium Prize problems (effectively impossible). The power law phenomenon manifests precisely in this intermediate difficulty range.
>
> It is not clear to us what a more compelling or more rigorous investigation would look like. If you have suggestions, we would greatly appreciate them!
>
> Thank you again for your insightful feedback, which will help strengthen both this work and our future research directions.

---

### Official Review · Reviewer_UNs5 · 2025-03-18

**Overall Recommendation:** 5

**Summary:**

This paper investigates the negative log of the average success rate scales as a power law with the number of attempts when LLMs make multiple independent attempts at a task (mathematical problems or jailbreaking). The authors identify a paradox that for any individual problem, success rates should improve exponentially (not as a power law) with more attempts. The paper resolves this paradox by demonstrating that power law scaling emerges from the distribution of per-problem single-attempt success probabilities. Specifically, the authors prove that a power law left tail in this distribution is necessary and sufficient for the emergence of aggregate power law scaling. The paper provides a theoretical framework that explains previously observed deviations from power law scaling and introduces a more sample-efficient method for estimating power law exponents.

**Claims And Evidence:**

- The claim that individual problems scale exponentially is supported by mathematical derivation in Section 2 and empirical evidence in Figure 3, showing negative log success rates falling exponentially for each problem.
- The necessary and sufficient conditions for power-law scaling are rigorously established through formal mathematical proofs (Theorems 3.1 and 3.2) and validated empirically.
- The explanation of why Llama 3 8B IT deviates from power law scaling (because its success distribution lacks the required heavy left tail) is empirically validated.

**Essential References Not Discussed:**

N/A

**Experimental Designs Or Analyses:**

- The authors appropriately visualize both per-problem exponential scaling and aggregate power law scaling.
- The distribution fitting and parameter estimation methods are appropriate.
- The backtesting approach for comparing estimators is rigorous, showing the distributional estimator achieves lower relative error.
- The authors appropriately account for sampling limitations and edge cases (problems with extremely low success probabilities).

**Methods And Evaluation Criteria:**

- The authors leverage existing datasets from prior work (Brown et al. 2024, Hughes et al. 2024), ensuring comparability with published results.
- The distributional models (Beta, Kumaraswamy, etc.) used to characterize success probability distributions are appropriate given the bounded nature of probabilities.
- he evaluation of the distributional estimator includes both agreement with least-squares on real data (Figure 6) and superior performance on synthetic data with known ground truth (Figure 7), providing a comprehensive analysis.

[1] Large Language Monkeys: Scaling Inference Compute with Repeated Sampling,  Brown et al., 2024

[2] Best-of-N Jailbreaking, Hughes et al., 2024

**Other Comments Or Suggestions:**

- Typo in line 255: "Kuamraswamy" -> "Kumaraswamy"

**Other Strengths And Weaknesses:**

**Weaknesses:**

- While the paper explains how power law scaling emerges, it offers limited insight into why single-attempt success rates have heavy-tailed distributions in the first place. The brief discussion of benchmark design and selection bias could be expanded.

- The empirical analyses are limited to specific model families and benchmarks. Verification across a broader range of models and tasks would strengthen generalizability.

**Questions For Authors:**

1. Your explanation focuses on the statistical properties of problem distributions that lead to power laws. Could you elaborate on potential causal factors that might create these heavy-tailed distributions in natural language tasks?
2. In Section 7, you speculate about connections to pretraining compute scaling laws. Have you found any empirical evidence that supports the "dark matter" hypothesis for neural scaling laws?

**Relation To Broader Scientific Literature:**

- This work extends recent work by Brown et al. (2024) on "Large Language Monkeys" and Hughes et al. (2024) on "Best-of-N Jailbreaking" by providing a theoretical explanation for their empirical findings.

- It connects to the broader literature on scaling laws in neural networks (Kaplan et al., 2020; Hoffmann et al., 2022) by revealing distributional foundations for observed scaling patterns.

[1] Training Compute-Optimal Large Language Models, Hoffmann et al., 2022

[2] Large Language Monkeys: Scaling Inference Compute with Repeated Sampling,  Brown et al., 2024

[3] Best-of-N Jailbreaking, Hughes et al., 2024

[4] Scaling Laws for Neural Language Models, Kaplan et al., 2020

**Theoretical Claims:**

I verified the key theoretical claims

- Theorem 3.1 (sufficiency): The proof correctly shows that power-law behavior near zero in the distribution yields aggregate power-law scaling.
- Theorem 3.2 (necessity): The proof correctly establishes that aggregate power law scaling requires a power-law left tail in the distribution.

---

> ### Author Rebuttal · Authors · 2025-03-31
>
> Thank you for your thorough and thoughtful review of our work. We will correct the typo you identified in line 255, changing "Kuamraswamy" to "Kumaraswamy." We address other points below:
>
> ### Origins of Heavy Left Tailed Distributions
>
> > The brief discussion of benchmark design and selection bias could be expanded.
>
> > Your explanation focuses on the statistical properties of problem distributions that lead to power laws. Could you elaborate on potential causal factors that might create these heavy-tailed distributions in natural language tasks?
>
> If our paper is accepted, we will use the additional page in our camera-ready version to expand on benchmark design and selection bias as factors leading to heavy-tailed distributions. Your question about causal factors creating these distributions touches on an important insight: power law scaling emerges in a "Goldilocks zone" of problem difficulty. For heavy left-tails to appear, we need problems that are challenging but not impossible—difficult enough to require many attempts yet still solvable. This explains why we wouldn't observe power law scaling when applying state-of-the-art models like GPT-4.5 or Claude 3.7 Sonnet to relatively simple benchmarks like GLUE (too easy), nor when applying these same models to extremely difficult tasks like Millennium Prize problems (effectively impossible). The power law phenomenon manifests precisely in this intermediate difficulty range.
>
> If you can think of a more rigorous way to investigate causal factors, we would welcome your suggestions!
>
> > The empirical analyses are limited to specific model families and benchmarks. Verification across a broader range of models and tasks would strengthen generalizability.
>
> We agree that verification across a wider range of models and tasks would strengthen generalizability. We relied on existing datasets from Brown et al. (2024) and Hughes et al. (2024) because generating 10,000+ attempts per model per problem involves substantial computational costs, e.g., drawing 10k attempts from 10 models across 5 benchmarks with 100 problems each would require 50 million samples. While computational constraints limited the scope of our current study, we view this as an important direction for future work and are exploring more efficient experimental designs to validate our theoretical framework more broadly.
>
> ### Dark Matter of Neural Scaling Laws
>
> Regarding your question about the "dark matter" of neural scaling laws, this is indeed the focus of our ongoing follow-up work! The experimental approach involves training numerous small models on scaling ladders and running scaling predictions in reverse to identify deviations from expected power law functional fits. This allows us to fit more complex functional forms and better understand deviations. We're particularly excited about this direction because experimenting with small models enables cheaper and faster iteration, but is currently not useful because extremely small models are poorly predictive of massive models. If we can figure out the appropriate scaling corrections, this would accelerate experimentation with larger models.
>
> Thank you again for your insightful comments and strong support for our work.

---

### Decision · Program_Chairs · 2025-05-01

**Decision:**

Accept (oral)

**Comment:**

The paper aims to understand scaling laws of the success rate when LLMs make multiple attempts to solve a problem through multiple trials. The work demonstrates why/when one observes a power-law (opposed to say, an exponential law) through a mix of theory and experiments and provide a sample efficient method to estimate the exponent.

All reviewers unanimously liked the submission due to its strong theoretical proofs (Theorems 3.1 and 3.2) and the rigorous design of the empirical evaluations. Moreover, reviewers also appreciated the explanations of cases in which power-laws are not observed (Llama 3 8B IT).

Reviewer 5fN8's concerns about a lack of deeper theoretical justifications on why power laws arise were cleared during the rebuttal phase.

As inference-time scaling is becoming more relevant to modern LLMs, this work is of high interest to the ICML audience. Given the high quality of the paper and overall interest in the work, I recommend acceptance.